## METHOD

# Multi-INTACT: integrative analysis of the genome, transcriptome, and proteome identifies causal mechanisms of complex traits

Jeffrey Okamoto[1*] , Xianyong Yin[1,2], Brady Ryan[1], Joshua Chiou[3], Francesca Luca[4], Roger Pique-Regi[4], Hae Kyung Im[5], Jean Morrison[1], Charles Burant[6], Eric B. Fauman[3], Markku Laakso[7], Michael Boehnke[1*] and Xiaoquan Wen[1*]

*Correspondence:
jokamoto@umich.edu;
boehnke@umich.edu;
xwen@umich.edu

[1] Department of Biostatistics and Center for Statistical Genetics, University of Michigan, Ann Arbor, MI 48109, USA
[2] Department of Epidemiology, School of Public Health, Nanjing Medical University, Nanjing, Jiangsu 211166, China
[3] Internal Medicine Research Unit, Pfizer Worldwide Research, Development and Medical, Cambridge, MA 02139, USA
[4] Center for Molecular Medicine and Genetics, Wayne State University, Detroit, MI 48201, USA
[5] Section of Genetic Medicine, Department of Medicine, University of Chicago, Chicago, IL 60637, USA
[6] Department of Internal Medicine, University of Michigan, Ann Arbor, MI 48109, USA
[7] Institute of Clinical Medicine, Internal Medicine, University of Eastern Finland and Kuopio University Hospital, Kuopio 70210, Finland

## Abstract

We present multi-integration of transcriptome-wide association studies and colocalization (Multi-INTACT), an algorithm that models multiple "gene products" (e.g., encoded RNA transcript and protein levels) to implicate causal genes and relevant gene products. In simulations, Multi-INTACT achieves higher power than existing methods, maintains calibrated false discovery rates, and detects the true causal gene product(s). We apply Multi-INTACT to GWAS on 1408 metabolites, integrating the GTEx expression and UK Biobank protein QTL datasets. Multi-INTACT infers 52 to 109% more metabolite causal genes than protein-alone or expression-alone analyses and indicates both gene products are relevant for most gene nominations.

## Background

Genome-wide association studies (GWAS) have greatly advanced our understanding of the genetic basis of complex diseases and traits by identifying numerous variant-level genetic associations. However, most identified associated variants lie in noncoding genome regions [1, 2], often obscuring their target genes and complicating the therapeutic target identification. Recent efforts to address this problem have resulted in the emergence of genome-scale molecular quantitative trait loci (QTL) annotations [3, 4] and the development of statistical methods bridging genetic variants with molecular phenotypes of gene candidates and complex traits [5–18]. These mechanism-aware putative causal gene (PCG) implication methods [8] focus on molecular phenotypes that can be unambiguously linked to specific genes. Henceforth, we refer to such molecular phenotypes as *gene products.* Key gene products commonly used in QTL mapping include transcriptome abundance, isoform usage, RNA decay rate, and protein abundance.

Existing mechanism-aware methods primarily implicate PCGs through a single gene product. Transcriptome-wide association studies (TWAS) [10–12, 14, 15] evaluate gene expression's mediating role by examining the correlation between genetically predicted expression with a GWAS trait. Colocalization analyses [14, 16, 17] aim to identify variant-level overlap of causal expression QTLs (eQTLs) and GWAS associations. Each of these methods has been shown to have distinct limitations due to statistical and biological factors such as horizontal pleiotropy and linkage disequilibrium (LD) hitchhiking, which result in false positives and negatives [5, 19–21].

While eQTL data have shown promise for expanding our understanding of complex trait genetic architecture [3, 22, 23], they do not always illuminate the true effects of causal genes on complex traits [24–26]. Recent work [27] suggests that most disease heritability cannot be explained by tissue-specific *cis*-expression quantitative trait loci (eQTL) data but rather by other molecular mechanisms. In particular, splicing QTLs (sQTLs) [3, 28] and protein QTLs (pQTLs) [25, 29] have shown to display minimal eQTL overlap and may independently influence disease heritability. Consequently, transcriptome-wide association studies (TWAS) and colocalization analysis of GWAS and eQTL data are often ineffective means of identifying causal genes. Numerous molecular phenotypes—including isoform usage, RNA degradation, and protein abundance—have demonstrated their relevance in explaining and predicting disease risk [30–43]. Joint analysis of multiple gene products can offer a holistic view of the underlying biology and improve the PCG implication performance.

In this study, we introduce Multi-INTACT, a mechanism-aware PCG inference method that aggregates colocalization and TWAS evidence across diverse gene products within a Bayesian framework. Multi-INTACT gauges the causal significance of a target gene concerning a complex trait and identifies the pivotal gene products. Using comprehensive simulations, we demonstrate the advantages of Multi-INTACT over existing methods. Finally, we use Multi-INTACT to detect PCGs influencing plasma metabolite levels via RNA transcript or protein levels.

## Results

### Method overview

The key idea of Multi-INTACT is to leverage information from *multiple* molecular phenotypes to implicate PCGs. Particularly, we define *gene products* as the molecular phenotypes that can be explicitly linked to genes. Contemporary experimental technology allows for the measurement of various gene products such as RNA abundance, isoform usage, RNA decay rate, and protein abundance. To examine potential gene-trait causative links, Multi-INTACT extends the canonical single-exposure (i.e., a single molecular phenotype) instrumental variables (IV) analysis/TWAS method to account for multiple endogenous variables, integrating colocalization evidence in the process. Genetic association analysis results of molecular QTLs and GWAS loci are essential to Multi-INTACT's inferential procedure. We summarize the Multi-INTACT method workflow in Fig. 1a.

While the proposed inference framework can be applied to incorporate many gene products simultaneously, for simplicity, we illustrate the method using two gene

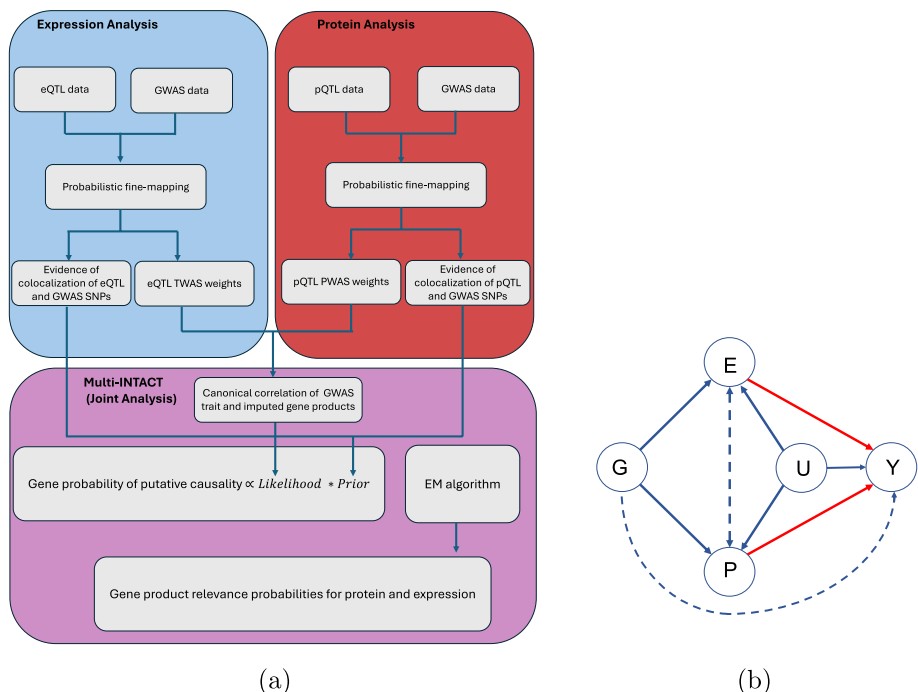

**Fig. 1 a** Multi-INTACT workflow. First, we perform multi-SNP fine-mapping analysis of eQTL, pQTL, and GWAS data (see Methods section). Then, we perform pairwise colocalization analysis of the eQTL and GWAS results, and separately, the pQTL and GWAS results. We generate eQTL TWAS weights and pQTL PWAS weights from the fine-mapping analysis. Using these weights, we compute the canonical correlation between the GWAS trait and imputed gene product levels. The canonical correlation and pairwise colocalization evidence serve as key inputs for Multi-INTACT analysis. Multi-INTACT utilizes an EM algorithm to determine the most relevant gene product based on the observed data. **b** Causal diagram that connects genetic variants *G*, expression levels of a candidate gene *E*, protein levels of a candidate gene *P*, and a complex trait *Y*. The node *U* represents latent confounders of effects between *E*, *P*, and *Y*. A similar diagram is assumed by multivariable Mendelian randomization methods. The edges that are highlighted in red are those that Multi-INTACT is designed to infer. The solid edges represent the graphical model assumed by most multivariable Mendelian randomization methods. The dashed lines emphasize that Multi-INTACT is designed to be robust to situations in which there are effects between gene products or there are violations of the exclusion restriction (i.e., direct-effect genetic variants)

products, gene expression levels (*E*) and protein abundance (*P*). The Multi-INTACT model is built upon the existing IV analysis framework, accommodating multiple endogenous variables (i.e., *E* and *P*) [44]. Figure 1b depicts the assumed directed acyclic graph (DAG) for potential relationships between molecular QTLs (*G*), gene products (*E* and *P*), unobserved confounding (*U*), and the complex trait of interest (*Y*). The primary goal of the statistical inference is to test for potential causal relationships from the gene products to the complex trait (i.e., $E \rightarrow Y$ and $P \rightarrow Y$). In the context of a causal model, these potential relationships are represented by the do-calculus [45] $P(Y \mid do(E))$ and $P(Y \mid do(P))$, while allowing flexible relationships between gene products. More specifically, we aim to answer the following two related scientific questions for each gene candidate:

1. Is the gene candidate a PCG, i.e., do *any* of its gene products exert a potential causal effect on the trait of interest?

2. Provided that the gene candidate is a PCG, what are the relevant gene products?

To assess the plausibility of a gene candidate being a PCG, Multi-INTACT applies an empirical Bayes procedure to compute a gene-specific posterior probability, denoted as the *gene probability of putative causality*. In particular, the likelihood computation generalizes from the existing single-trait TWAS methodology by constructing the composite instrument variables, $\hat{E}$ and $\hat{P}$, using their respective molecular QTLs and subsequently testing their *canonical correlation* with $Y$. These composite IVs are naturally interpreted as the genetic prediction of the corresponding molecular phenotypes, and the procedure follows the principles of the IV analysis framework [46, 47]. The prior formulation is designed to ensure the key causal assumptions and validate the putative causality claim. Specifically, the prior incorporates the colocalization evidence of molecular QTLs and GWAS hits to guard against violations of the exclusion restriction (ER) assumption caused by widespread LD in genetic data. Similar to the original INTACT method [8], the incorporation of probabilistic colocalization evidence in the Multi-INTACT model relies on Bayes' rule and is justified by Dempster's rule of combination [48]. We note that under various possible causal scenarios (Additional file 1: Fig. S1), not all types of molecular QTLs are necessarily colocalized with GWAS hits. Additionally, considering the low practical power to detect colocalization [19], our implementation requires at least one type of molecular QTL to exhibit modest colocalization evidence, i.e., has a gene-level colocalization probability greater than zero, as a minimum necessary condition for a candidate gene to be classified as a PCG. Finally, the complement of a gene probability of putative causality is a local false discovery rate (lfdr), which is suitable for the standard Bayesian false discovery rate (FDR) control procedure to guard against type I errors [49–54].

To address the second question on identifying relevant gene products, we formulate a model selection problem considering all possible causal relationships from examined gene products to the complex trait. In our illustrative example, Multi-INTACT evaluates the posterior probabilities for the following four mutually exclusive models:

1. $M_0$: neither $E$ or $P$ exerts an effect on $Y$, i.e., the null model
2. $M_E$: only $E$ exerts an effect on $Y$, i.e., the $E \rightarrow Y$ model
3. $M_P$: only $P$ exerts an effect on $Y$, i.e., the $P \rightarrow Y$ model
4. $M_{E+P}$: both $P$ and $E$ exert effects on $Y$, i.e., the $(E, P) \rightarrow Y$ model

For a general case where there are $p$ gene products considered, there are $2^p$ models to be compared. This presents a practical limitation for Multi-INTACT, as the number of models increases rapidly with $p$.

Multi-INTACT implements an EM algorithm to estimate the empirical Bayes prior distribution among the three competing alternative models by pooling information across all gene candidates. It subsequently evaluates the posterior model probabilities, i.e., $\Pr(M_E \mid \text{data})$, $\Pr(M_P \mid \text{data})$, and $\Pr(M_{E+P} \mid \text{data})$, for each gene candidate using the Bayes rule. Finally, Multi-INTACT reports a *gene product relevance probability* of each gene product for a given candidate by marginalizing the corresponding posterior model probabilities:

$$\begin{aligned}
\Pr(E \text{ exerts an effect on } Y \,|\, \text{data}) &= \Pr(M_E | \text{data}) + \Pr(M_{E+P} \,|\, \text{data}) \\
\Pr(P \text{ exerts an effect on } Y \,|\, \text{data}) &= \Pr(M_P \,|\, \text{data}) + \Pr(M_{E+P} \,|\, \text{data}).
\end{aligned} \tag{1}$$

We provide technical details and features of the Multi-INTACT method in Methods section. In brief, the Multi-INTACT model can be represented by a structural equation model (SEM) that allows pleiotropic effects. Hence, it is robust against some of the most common violations of the ER assumptions using genetic data. Importantly, although the causality claim is derived based on the one-sample design (i.e., all genetic, molecular phenotypes, and complex traits are measured on a single cohort), it can be extended to multi-sample designs, which are common among available genomic and genetic data.

**Simulation study**

To evaluate the performance of Multi-INTACT, we perform extensive simulation studies based on genetic data from GTEx. We extend the simulation design introduced in [8], which uses real genotypes of 477K SNPs on chromosome 5 from 500 GTEx samples. The selected genomic region contains 1198 consecutive genes, each with at least 1500 common *cis*-SNPs, some of which are located in the overlapping cis-regions of multiple gene candidates. We consider a multi-sample design to simulate the molecular and GWAS phenotypes, where the residuals of $E$, $P$, and $Y$ are uncorrelated after controlling for their shared genetic components. The phenotype data are generated by randomly sampling the DAGs shown in the second row of Additional file 1: Table S1. Note that the data generative models differ from the Multi-INTACT model, as they make additional assumptions connecting $G$, $E$, and $P$. Each assembled simulated dataset contains 1198 genes with $\sim 80\%$ non-PCGs and $\sim 20\%$ PCGs. The causal mechanism of each simulated PCG follows a discrete distribution of $M_E, M_P$, and $M_{P+E}$ models, which is varied across different datasets. In total, we generate 100 datasets for analysis. Complete simulation details are provided in Methods section and Additional file 1: Supplemental Methods.

For each simulated dataset, we perform multi-SNP fine-mapping analyses using individual-level genotype-phenotype data for all molecular and complex traits. We then separately conduct colocalization [14] and TWAS [15] analysis for the protein-GWAS and expression-GWAS data. The resulting single-molecular trait integrative analysis data are subsequently used as input for Multi-INTACT. For comparison, we also perform single-molecular trait INTACT [8] analysis for expression and protein data, respectively.

We first assess the ability of the Multi-INTACT method to identify PCGs. Specifically, we evaluate the averaged power and false discovery rate at the target FDR control level of 5% across all simulated datasets for all methods. The results show that Multi-INTACT exhibits optimal power while properly controlling type I errors (Fig. 2). Multi-INTACT outperforms existing methods that use a single gene product for PCG implication, including TWAS, colocalization analysis, and INTACT. We find that TWAS methods suffer from severely inflated type I errors due to failure to account for LD hitchhiking, while colocalization methods properly control type I errors but are overly conservative. Finally, while the single-trait INTACT method maintains fairly consistent power and FDR across TWAS prediction models (Additional file 1: Table S2), Multi-INTACT achieves higher power mainly because additional gene products are considered.

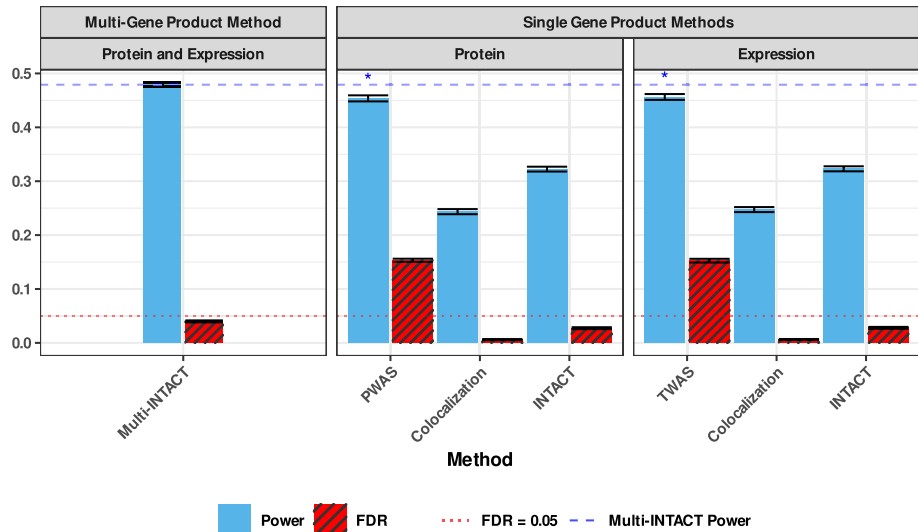

**Fig. 2** Average realized power and false discovery rates for each integrative PCG-implicating method at the 5% level over 100 simulated data sets. Methods are grouped by whether they consider multiple molecular phenotypes. Methods that consider only one molecular phenotype are grouped by omics platform. For ease of comparison, we include a dashed blue line to denote the Multi-INTACT power. Power columns representing methods with excessive false discoveries are marked with an asterisk. Error bars represent the standard error of the mean

To illustrate Multi-INTACT's ability to accommodate results from different TWAS prediction methods, we run Multi-INTACT using various popular molecular phenotype prediction models and observe that Multi-INTACT's power and FDR results remain fairly consistent across different TWAS models (Additional file 1: Table S3).

Additionally, we compute the gene probabilities of putative causality using only the summary association statistics of the simulated complex trait. We find the corresponding results are nearly identical to those obtained from individual-level GWAS data (Additional file 1: Table S4). It should be acknowledged that our summary-level statistics represent a best-case scenario, as the LD information perfectly matches the underlying GWAS samples. In practice, LD information derived from a population reference panel often leads to imperfect characterization of sample LD and less-accurate inference results than what are obtained in this experiment.

Next, we evaluate Multi-INTACT as a means to identify relevant gene products for PCGs. To this end, we first apply the EM algorithm to estimate the proportions of PCG mechanisms in each simulated dataset. Then, we compute the corresponding gene-level posterior model probabilities for $M_E, M_P$, and $M_{E+P}$. We find that the EM algorithm estimates of the mechanism distributions are reasonably accurate, i.e., the true proportion of PCGs following a mechanism always falls within the interquartile range of the EM algorithm estimate distribution (Additional file 1: Fig. S2). The distributions of the posterior model probabilities, stratified by the underlying true causal mechanisms, are shown in Fig. 3 and Additional file 1: Figs. S3–S5. Although the data do not completely distinguish the true mechanism in all settings, we find that the true mechanisms are always assessed with the highest posterior probabilities on average. Finally, we compute the gene product relevance probabilities for $E$ and $P$ across all genes. Using the gene product relevance probabilities as

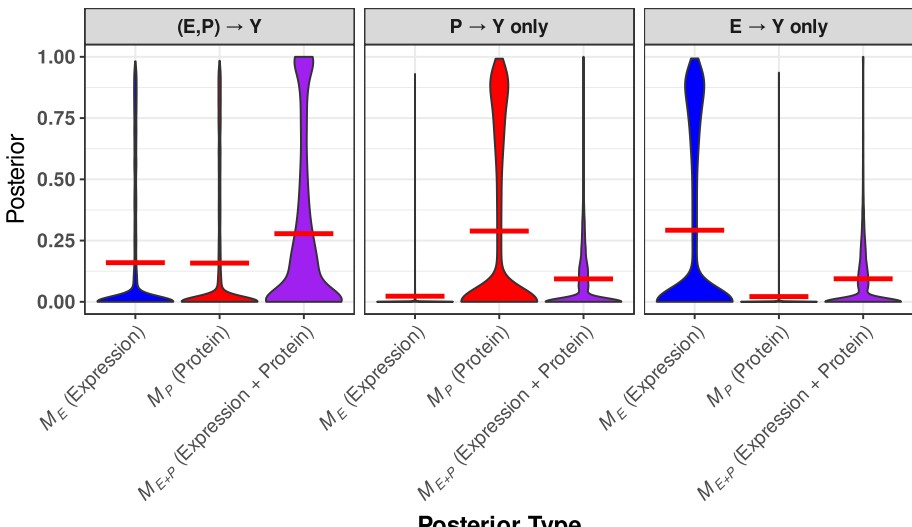

**Fig. 3** Distributions of posterior probabilities for each gene product-to-trait effect scenario and three posterior probability types. For example, for genes represented in the left-most panel, both expression and protein levels have direct effects on the complex trait. The left-most violin plot represents the distribution of posterior probabilities of the model in which only expression has a direct effect. The distributions represent genes that have nonzero causal effects on Y (through at least one of expression or protein). For each violin plot, a horizontal red line denotes the mean of the distribution

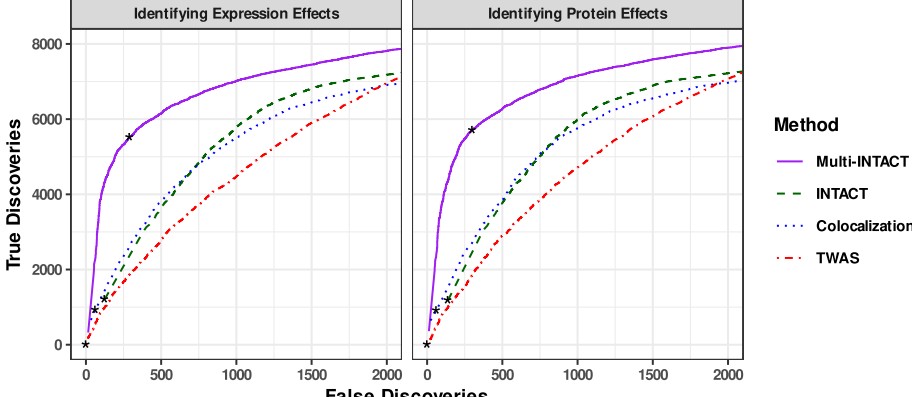

**Fig. 4** True discovery versus false discovery curves comparing the performances of Multi-INTACT and INTACT for classifying gene-to-trait effects. Curves are truncated at a false discovery rate threshold of 0.20 for Multi-INTACT in order to represent the highest-ranked genes. Results represent 100 simulated data sets with causal genes from a variety of possible DAGs. For Multi-INTACT, we use gene product relevance probabilities $P(E \mid \text{data})$ and $P(P \mid \text{data})$ as scores for detecting $E \rightarrow Y$ and $P \rightarrow Y$ effects, respectively. We use the INTACT posterior, fastENLOC gene-level colocalization probability, and TWAS z-score magnitude with the respective molecular phenotype (expression, left; protein, right) as a classification score in each panel for comparison. For each curve, the false discovery rate threshold of 0.05 is denoted by an asterisk

classification scores, we generate cumulative true discovery versus false discovery curves (Fig. 4). Multi-INTACT outperforms alternative methods that rely on a single molecular trait at a time in identifying relevant causal gene products from highly ranked gene candidates.

### Analysis of METSIM Metabolon metabolite GWAS data

To demonstrate the advantages of jointly considering multiple gene product molecular phenotypes, we apply INTACT and the proposed Multi-INTACT procedures to 1408 Metabolic Syndrome in Men (METSIM) study plasma metabolite GWASs [55], integrating the UK Biobank pQTL data [4] and multi-tissue GTEx eQTL data (v.8) [3] to identify PCGs and putative biological mechanisms. Details of the pre-processing analysis, including algorithms used for TWAS and colocalization, are discussed in Methods section.

The METSIM Metabolon metabolite study includes 10,188 Finnish men from Kuopio examined from 2005 to 2010 [55]. Study participants are whole-genome sequenced, yielding >26 million represented variants that pass QC procedures. For a description of the METSIM data pre-processing, including QC, genetic association analysis, and multi-SNP fine-mapping analysis, refer to Additional file 1: Supplemental Methods.

Our previous work highlights the discordance, or the lack of inferential reproducibility, in implicating PCGs when colocalization and TWAS analyses are applied to gene expression data [8, 20]. To determine whether a similar pattern holds in applications of proteomics data, we first perform UK Biobank pQTL-metabolite GWAS integrative analysis and compare the implicated PCGs (at 5% FDR level) from proteome-wide association study (PWAS), colocalization, and INTACT analyses across the 1408 metabolites. PWAS identifies 2217 genes, colocalization analysis identifies 170 genes, and INTACT, combining colocalization and PWAS evidence, identifies 293 PCGs. Our results imply that ~95% of the PWAS genes do not show colocalization evidence, suggesting that most of these findings are likely due to the LD hitchhiking effects previously discussed in the context of TWAS analysis. Although INTACT is effective at guarding against LD hitchhiking, its statistical power is compromised and can be improved by incorporating more relevant gene products.

Next, we apply Multi-INTACT, integrating the UK Biobank pQTL data and the eQTL data representing one (at a time) of 49 tissues from the GTEx project. We first compute the gene probabilities of putative causality and identify PCGs at 5% FDR level separately in each metabolite-tissue pair. The full Multi-INTACT results from this analysis are summarized in the supplemental data. Among the tested 68,992 tissue-metabolite pairs, Multi-INTACT identifies 8610 PCG-tissue-metabolite triplets, notably more than the expression-only or protein-only INTACT analyses which implicate 4128 and 5682 triplets, respectively. Upon stratifying the results by tissues, it is clear that although the number of discoveries varies by tissue, Multi-INTACT consistently implicates more genes than the expression-only and protein-only INTACT analyses (Fig. 5). The increase in the discoveries illustrates the improved power of combining relevant gene products. Additionally, while a large proportion of the triplets identified by Multi-INTACT are also implicated by at least one of the INTACT analyses, there are many PCGs identified only by Multi-INTACT (Additional file 1: Fig. S6).

The overlap of PCG-metabolite pairs identified between tissues varies widely across the 49 GTEx tissues (Fig. 6a). Unsurprisingly, Multi-INTACT results derived from brain cerebellum and cerebellar hemisphere expression data share a high proportion of identified PCG-metabolite pairs. Of the 894 unique PCG-metabolite pairs discovered in at least one tissue, we find that 23% of the PCG-metabolite pairs are implicated in a single tissue, and 50% of pairs are implicated in between 2 and 10 tissues (Fig. 6b). Meanwhile,

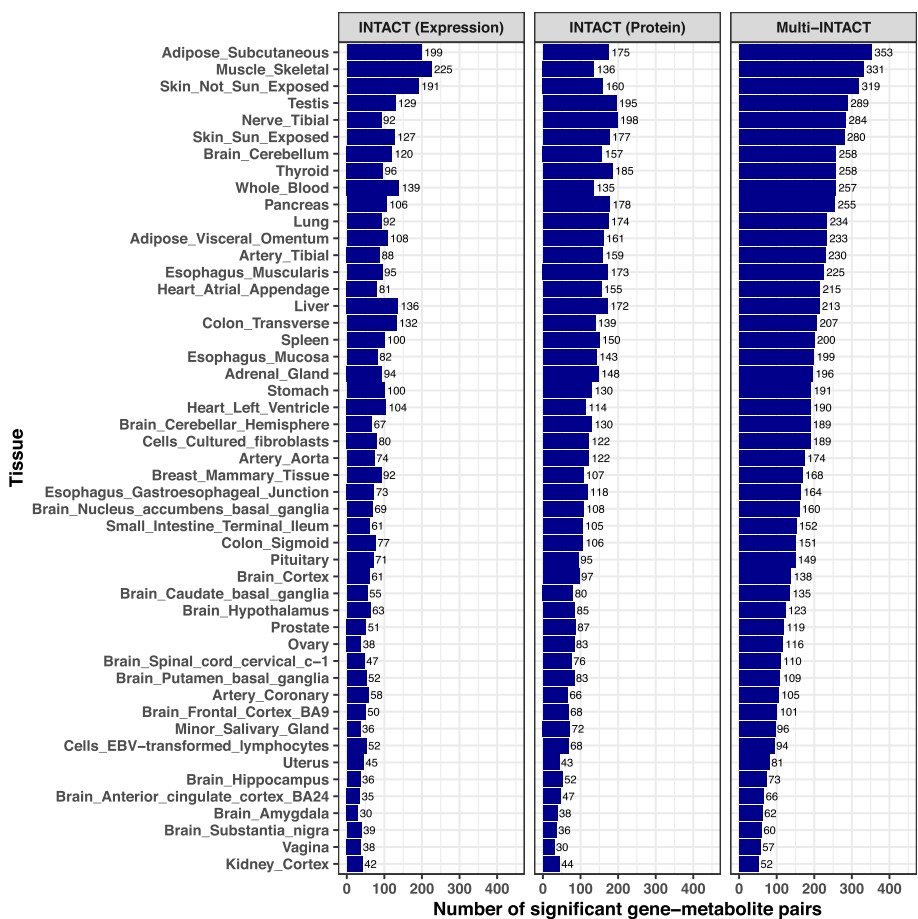

**Fig. 5** Multi-INTACT PCG implication results summary, by tissue. For each tissue-specific analysis, only genes with both expression and protein data are tested

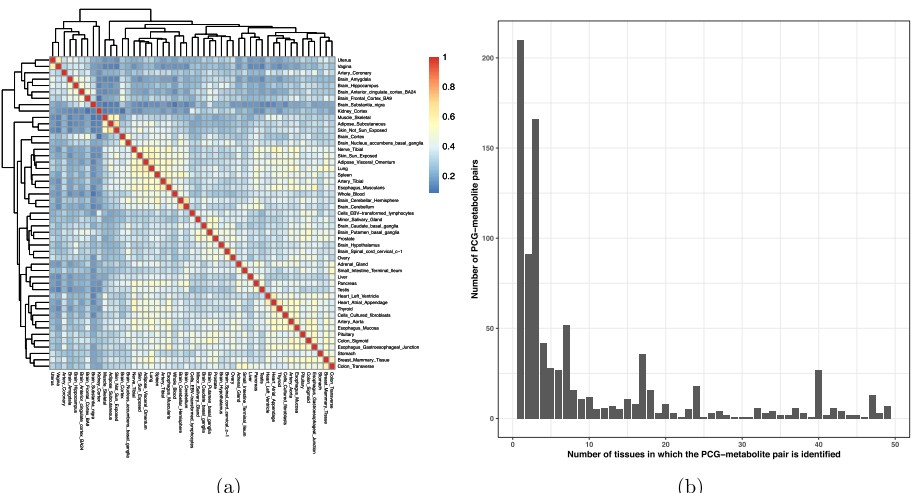

(a)    (b)

**Fig. 6** Overlap between gene-metabolite pairs discovered by Multi-INTACT using expression data across GTEx tissues. **a** Pairwise overlap of PCG-metabolite pairs implicated across tissues. Fill color denotes (# gene-metabolite pairs implicated by both tissues)/(# gene-metabolite pairs implicated by at least one of the tissues). **b** The distribution of the number of tissues in which Multi-INTACT identifies gene-metabolite pairs

4% of PCG-metabolite pairs are identified in >40 tissues. We find that the discrepancy of PCG discovery across tissues is driven by the tissue-dependent variability of eQTL discovery, which has been observed in tissue-specific TWAS analysis and attributed to both variation in statistical power and to biological factors, such as tissue-specific gene regulation. As a result, many genes do not have expression prediction models in all tissues. Here, we caution against interpreting tissue-specific PCG discoveries solely by biological (or statistical) factors.

To validate the Multi-INTACT PCG findings, we compare our inferences to a high-quality annotated causal gene set for a group of metabolites by a knowledge-based approach (KBA). The KBA nominates PCGs by matching known metabolite biochemistry to functions of genes near strong GWAS signals [40, 41, 56, 57]. For our validation analysis, we limit the KBA nominations to genes that have both pQTLs and eQTLs in at least one tissue (i.e., candidates for Multi-INTACT analysis), where the KBA nominates 423 unique gene-metabolite pairs in total. The overlapping of the PCGs implicated by each integrative approach and the KBA are summarized in Fig. 7. The expression-only and protein-only analyses identify 238 and 184 known gene-metabolite KBA pairs, respectively. In contrast, Multi-INTACT results overlap with 304, or ~70%, of

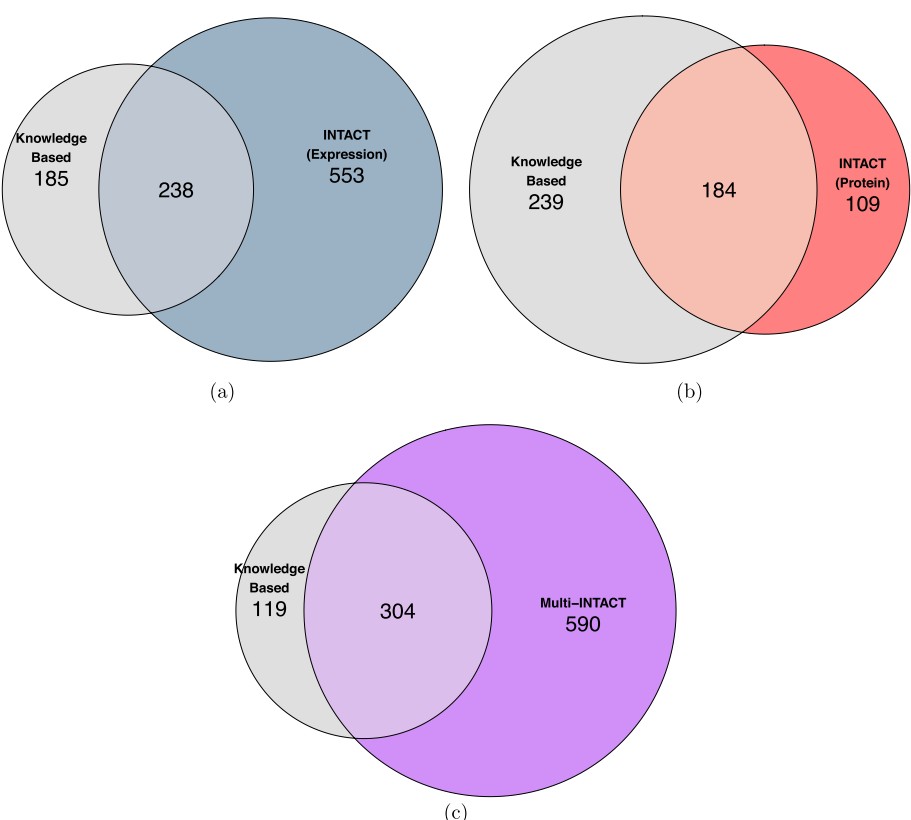

(a)                                                                                      (b)

(c)

**Fig. 7** Overlap between gene-metabolite pairs discovered by integrative approaches and the knowledge-based approach (KBA). **a** INTACT using GTEx multi-tissue eQTL data. Numbers represent pairs implicated in at least one tissue. **b** INTACT using the UK Biobank pQTL data set. **c** Multi-INTACT using both GTEx and UK Biobank QTL data sets. There are 289 unique pairs implicated by KBA and either INTACT-expression or INTACT-protein. Of these pairs, 287 are represented in the intersection of the KBA and Multi-INTACT discoveries

the annotated KBA pairs, demonstrating superior power over the existing methods. Although the KBA is intended to benchmark Multi-INTACT, it does not provide an exhaustive list of all biologically-feasible PCG-metabolite pairs. Some pairs implicated by Multi-INTACT, but not the KBA, may reflect known biology. For example, we identify *GSTA1* as a PCG for DHEA-S (C100000792) in both liver and adrenal cortex tissues. This finding may reflect previous biological evidence that both *GSTA1* and DHEA-S have roles in metabolizing prostaglandins [58, 59].

Next, we investigate the directional consistency of different gene product-to-trait effects from the PCGs implicated by Multi-INTACT. The central dogma states that the flow of genetic information from DNA to RNA to protein is one-directional, implying that the sign of a gene's effect should be the same across gene products. Despite this, many studies [60–63] report complex relationships between expression levels, protein abundance, and disease risks. Although Multi-INTACT is not specifically designed to estimate gene product-to-trait effects, the signed TWAS and PWAS test statistics from *confidently inferred PCGs* should represent their qualitative directional effects. PCGs are confidently inferred if they are statistically significant based on gene probability of putative causality at 5% FDR level. In this analysis, we further select a subset of Multi-INTACT gene-tissue-metabolite triplets for which both gene expression and protein abundance are deemed relevant gene products. To this end, we focus on the set of triplets whose gene product relevance probabilities for both expression and protein are $\geq 0.50$. For this selected set of triplets, we examine the directional consistency of the *z*-statistics from the corresponding TWAS and PWAS analyses.

Overall, of the 8007 analyzed triplets, 5000 ($\sim 62\%$) show matching directions from both gene products, while 3007 ($\sim 38\%$) show opposite directions. Interestingly, some tissues known to play key roles in metabolism show high directional consistency. For example, the highest directional-consistency proportion is observed in liver (Fig. 8), where 88% expression-to-trait and protein-to-trait effects are concordant and significantly higher than the remaining tissue "population" ($p$ value $= 5.2 \times 10^{-12}$). The concordant proportion increases to 96% when intersecting the Multi-INTACT results with the KBA results. Finally, we compute the Spearman correlation coefficient to compare the signed negative log $p$ values of the TWAS and PWAS analyses. For the triplets implicated by Multi-INTACT, the correlation estimate (0.319, $p$ value $= 2.18 \times 10^{-188}$) is higher than the "population" average for tested metabolite-tissue pairs (mean $= 0.156$, variance $= 0.002$). A recent study comparing PWAS and TWAS in four blood lipid traits reports a similar range of correlation estimates (0.083–0.144) as our population average [64]. Following the sign concordance analysis, we investigated the overlap of significant PWAS and TWAS results and found it to be substantial (Additional file 1: Fig. S7).

Lastly, we demonstrate the utility of Multi-INTACT for gene set enrichment analysis (GSEA). We focus our analysis on Multi-INTACT results derived from lipid metabolite data and liver expression data based on known biology [65] and results from previous analyses. For each target gene tested among the liver-lipid metabolite results, we form an aggregated probability of putative causality by combining the gene probability of putative causality across metabolites (see Methods section for details). We then apply INTACT-GSE [8], a recently introduced GSEA method. We use the aggregated probability of putative causality as input for INTACT-GSE. These probabilities have the

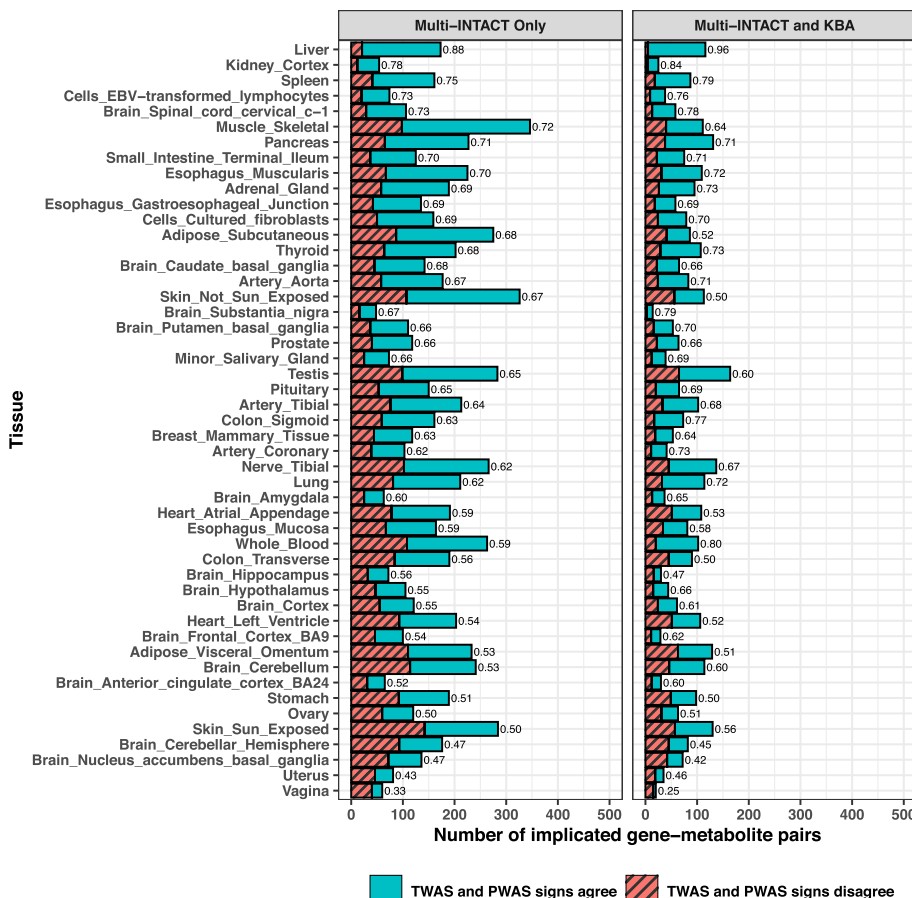

**Fig. 8** Gene-metabolite pairs with Multi-INTACT gene product relevance probabilities greater than 0.5 for both expression and protein, by tissue, with proportions of PWAS and TWAS test statistics that were the same/opposite signs. In the right panel, we show the intersection of genes implicated by Multi-INTACT and the knowledge-based approach (KBA) results. For each tissue-specific analysis, only genes with both expression and protein data are tested

unique advantage of quantifying the uncertainty of the presence/absence of a PCG for at least one lipid metabolite, which is a key to obtaining unbiased enrichment estimates for candidate gene sets. We examine 213 Biological Process (BP) GO terms, estimating enrichment and the corresponding 95% confidence intervals. In summary, we identify 16 GO terms that are nominally significant at the 5% level, i.e., their confidence intervals do not overlap with 0 (Table 1). Among these findings, cholesterol metabolic process (GO:0008203), steroid metabolic process (GO:0008202), and chemical reactions and pathways involving fatty acids (GO:0006631) are strongly enriched and reflect the well-known roles of liver in human metabolism. Full INTACT-GSE results are available in the supplemental data.

## Discussion

We present Multi-INTACT, a novel statistical method for mechanism-aware PCG implication by integrating GWAS and multiple types of molecular QTL data. Through simulations and real data analysis, we demonstrate that Multi-INTACT properly controls type

**Table 1** Top enriched Biological Process GO terms from the INTACT-GSE gene set enrichment analysis

| Biological Process GO term ID | Term | INTACT-GSE enrichment estimate (log odds ratio) | 95% confidence interval |
|---|---|---|---|
| GO:0008203 | Cholesterol metabolic process | 3.069 | (1.907, 4.231) |
| GO:0008202 | Steroid metabolic process | 2.633 | (1.609, 3.657) |
| GO:0006644 | Phospholipid metabolic process | 2.141 | (1.057, 3.225) |
| GO:0006629 | Lipid metabolic process | 2.026 | (1.045, 3.007) |
| GO:0042632 | Cholesterol homeostasis | 2.018 | (0.840, 3.197) |
| GO:0006631 | Fatty acid metabolic process | 1.997 | (0.931, 3.064) |
| GO:0006082 | Organic acid metabolic process | 1.978 | (1.025, 2.930) |
| GO:0016042 | Lipid catabolic process | 1.913 | (0.854, 2.972) |
| GO:0055088 | Lipid homeostasis | 1.894 | (0.737, 3.051) |
| GO:0046395 | Carboxylic acid catabolic process | 1.893 | (0.842, 2.943) |
| GO:0009636 | Response to toxic substance | 1.852 | (0.715, 2.989) |
| GO:0006805 | Xenobiotic metabolic process | 1.805 | (0.520, 3.091) |
| GO:0001889 | Liver development | 1.616 | (0.333, 2.899) |
| GO:0007041 | Lysosomal transport | 1.616 | (0.333, 2.899) |
| GO:0007034 | Vacuolar transport | 1.485 | (0.223, 2.747) |
| GO:0042594 | Response to starvation | 1.373 | (0.124, 2.621) |

Each listed term is nominally significant at the 5% level

I errors by effectively guarding against LD hitchhiking and identifies causal genes with increased statistical power compared to other popular PCG implication methods. Additionally, its ability to simultaneously evaluate the relevance of multiple gene products (i.e., those exerting direct effects on complex traits) can help gain insights on detailed molecular mechanisms of complex diseases, serving as a potential tool for drug target discovery.

We motivate and derive the Multi-INTACT method under a non-parametric graphical model representing the established IV analysis with multiple endogenous variables, where the model assumptions focus on the conditional independence relationships rather than the distributional assumptions on molecular and complex phenotypes. Specifically, the graphical model does not assume normality of each phenotype or additivity (see the SEM in Methods section). Consequently, Multi-INTACT is robust and capable of handling various types of (e.g., quantitative, binary, and categorical) phenotypic data. Furthermore, we show that with additional linearity and normality assumptions, the Multi-INTACT model can be represented by a structural equation model (Eq. 2 in Methods section), which naturally extends the SEMs previously used in integrative genetic association analysis with a single molecular phenotype [5, 6].

The unique inference strategy implemented in Multi-INTACT is to leverage colocalization evidence and prevent TWAS/PWAS association signals driven *solely* by pleiotropic effects. This is a key difference from the alternative strategy that attempts to explicitly estimate and control potential pleiotropic effects [5, 6]. In our numerical experiments and real data analysis, we find that the Multi-INTACT strategy is conceptually simple, computationally efficient, and practically effective. Nevertheless, we acknowledge that, despite many unresolved statistical and computational challenges, accurate estimation of pleiotropic effects has the potential to further increase the power

of PCG implication. We will explore the possibility of combining these strategies in future work. Additionally, the use of a multiplicative combination of colocalization and multivariable regression evidence (via Bayes rule) in Multi-INTACT inference can also be justified by Dempster-Shafer (DS) theory [48] as described in the original INTACT paper. Specifically, the combination of the colocalization and regression evidence is an application of Dempster's rule of combination. This application is a generalization of the INTACT evidence integration step in which each evidence source integrates information across multiple gene products rather than only the transcriptome.

In our joint analysis of METSIM Metabolon metabolite GWAS, GTEx eQTL, and UK Biobank pQTL data, we highlight the practical utility of Multi-INTACT, showing its potential in validating known and uncovering unknown molecular mechanisms of complex traits. Our investigation of directional consistency of gene-to-trait effects of PCGs between different gene products illustrates the complexity of the underlying scientific problem: although the observed patterns are sensible and expected by existing theoretical and experimental evidence, it is considerably challenging to interpret these findings for individual gene-tissue pairs. The observational association data have intrinsic limitations for further explorations. We hope that our findings can serve as a starting point for careful design of experimental investigations. In addition, we demonstrate that Multi-INTACT output can be directly applied to gene set enrichment analysis, a key feature to further validate and explore PCG findings. With improved information from multiple gene products, gene set enrichment analysis becomes more powerful compared to our previous analysis using a single gene product [8].

Although Multi-INTACT offers multiple improvements over existing PCG implication techniques, it does have limitations. The input from the existing colocalization and TWAS/PWAS analysis methods and the quality of currently available genetic data ultimately impact the performance of the Multi-INTACT analysis. This is clearly demonstrated from our simulation analysis (Additional file 1: Fig. S8). Future advances in methods development and data generation in the related areas will help improve Multi-INTACT analysis. Other limitations that Multi-INTACT shares with many integrative analysis methods include a reliance on prior biological knowledge to determine relevant tissues or cell types and a focus on model selection/testing rather than estimating gene-to-trait effects. Other studies [66, 67] have developed algorithms that attempt to infer the causal tissue, but we are not aware of any such method that considers multiple gene products. Nevertheless, these may be able to complement Multi-INTACT in an integrative analysis. Finally, Multi-INTACT explicitly focuses on gene products. Although this feature makes implicating PCGs more direct and interpretable, there is room for improvement by incorporating some indirect but potentially important genomic information (e.g., chromatin structure, methylation status, and 3D genome structure). Consideration of output from methods that consider these datatypes such as the ABC model [68] or PUMICE [69] may increase the quality of our PCG implications. Under the Bayesian framework of Multi-INTACT, it seems feasible to integrate this additional genomic information by modifying the prior formulation. Another promising extension of Multi-INTACT would be to account for a multi-ancestry study design. Previous studies have shown that incorporating ancestry into TWAS prediction models can yield higher predictive accuracy of expression prediction models and increased efficiency in

detecting gene-trait associations [70, 71]. We suspect that the approach to constructing race-stratified molecular gene product prediction models described in [70] could be adapted to improve the performance of Multi-INTACT. Additionally, if the GWAS data contains admixed populations, our methods could borrow ideas from existing work [71] which consider global genetic ancestry in order to mitigate false positives. We will address these challenges in our future work.

## Conclusion

In conclusion, Multi-INTACT integrates multiple gene products to reliably infer causal genes and mechanisms. Importantly, Multi-INTACT assesses which molecular gene products are the most relevant to disease, providing insights into possible drug targets. The Multi-INTACT software implementation is computationally efficient and can be applied using summary statistics of GWAS data. While we show that Multi-INTACT is useful for integrating expression and protein QTL datasets, our method has strong potential for studying complex trait etiology as the availability of molecular QTL datasets increases.

## Methods

### Structural equation model for Multi-INTACT

The Multi-INTACT method can be represented by a structural equation model (SEM), extending the SEM for TWAS analysis [8]. Consider a sample of $N$ individuals from a single cohort. For a target gene, let $E(N \times 1), P(N \times 1)$, and $G(N \times p)$ denote their expression levels, protein abundance, and genotypes for $p$ *cis* genetic variants, respectively. The measurements of the complex trait of interest and the unobserved confounding are represented by $Y(N \times 1)$ and $U(N \times 1)$. All observed phenotype measurements are assumed to be pre-centered. Let $p$-vectors $\boldsymbol{\beta}_E$ and $\boldsymbol{\beta}_p$ denote the genetic effects on respective gene products for all *cis*-variants of the target gene and the $p$-vector $\boldsymbol{\beta}_Y$ represents potential pleiotropic effects. Scalars $\theta_E$ and $\theta_P$ denote effects of unobserved confounding on expression and protein levels, respectively. Finally, we denote the $E \rightarrow Y$ and $P \rightarrow Y$ effects by $\gamma$ and $\delta$, which are of interest for inference. Reflecting the graphical model in Fig. 1b, the proposed SEM is given by

$$
\begin{aligned}
&E = G\boldsymbol{\beta}_E + \theta_E U + e_E, \;\; e_E \sim N(\mathbf{0}, \sigma_E^2 I) \\
&P = G\boldsymbol{\beta}_P + \theta_P U + e_P, \;\; e_P \sim N(\mathbf{0}, \sigma_P^2 I) \\
&Y = G\boldsymbol{\beta}_Y + \gamma E + \delta P + \theta_Y U + e_Y, \;\; e_Y \sim N(\mathbf{0}, \sigma_Y^2 I).
\end{aligned}
\tag{2}
$$

Noticeaby, our model does not assume that $E$ and $P$ have separate effects on Y. Specifically, the Multi-INTACT model, represented by the SEM, is a conditional model only focusing on the potential causal relationships $E \rightarrow Y$ and $P \rightarrow Y$. It does not explicitly specifying other possible causal relationships (e.g., between $E$ and $P$). Implicitly, potential $E \rightarrow P$ and/or $P \rightarrow E$ effects effects are accounted for by the corresponding residual error terms, i.e., $e_E$ and $e_P$. Alternatively, this approach can be understood as marginalizing all over other causal relationships between variables, thus representing all graphical models in Additional file 1: Table S1. The SEM is also similar to the multivariable Mendelian

randomization (MVMR) models discussed in genetic epidemiology [44, 72–74]. Their major difference lies in the inference strategy.

To assess a target gene for its putative causality, we consider testing a null hypothesis,

$$M_0 : \gamma = 0 \text{ and } \delta = 0. \tag{3}$$

In Multi-INTACT, we adopt the Bayesian strategy of model selection and assess the posterior probability of $M_0$. The strategy also naturally extends to the subsequent task of assessing relevant gene products by evaluating (and marginalizing from) the posterior probabilities of the alternative models,

$$\begin{aligned} M_E &: \gamma \neq 0 \text{ and } \delta = 0 \\ M_P &: \gamma = 0 \text{ and } \delta \neq 0 \\ M_{E+P} &: \gamma \neq 0 \text{ and } \delta \neq 0. \end{aligned} \tag{4}$$

Note that the main goal of Multi-INTACT is to distinguish between different models representing different mechanisms for a candidate gene. These models are characterized by $M_0, M_E, M_P$, and $M_{E+P}$. Our goal is fundamentally different from rigorously estimating the causal effects $\gamma$ and $\delta$. This important point dictates our formulation of the statistical problems and the inference procedure for fitting SEM (2).

A key feature of the Multi-INTACT method is the use of colocalization evidence to guard against widespread LD hitchhiking. This feature is motivated by the following observation from the proposed SEM: if a gene product has a non-zero effect on the complex trait of interest (e.g., $\delta \neq 0$), its causal molecular QTL must also impose a non-zero genetic effect on the complex trait (e.g., $\beta_p \cdot \delta \neq 0$). That is, colocalization is a necessary condition for $\gamma \neq 0$ or $\delta \neq 0$ under the proposed model. In comparison, TWAS and PWAS signals driven by LD hitchhiking are *not* expected to exhibit colocalization evidence. However, in current practice, colocalization analysis is often severely underpowered [19]. Acknowledging this caveat, instead of requiring all implicated PCGs to show a high level of colocalization evidence, our default implementation of Multi-INTACT essentially filters out candidate genes that lack even modest colocalization evidence.

To compute the likelihood, we simplify SEM Eq. 2 for inference. Because the confounding $\boldsymbol{U}$ is unobserved, its effects on $\boldsymbol{E}$, $\boldsymbol{P}$, and $\boldsymbol{Y}$ are absorbed into the respective residual error terms. Consequently, $\boldsymbol{E}$ and $\boldsymbol{P}$ become endogenous variables (under the one-sample design) in the final regression equation of $\boldsymbol{Y}$. To properly examine the gene-to-trait effects $\gamma$ and $\delta$, Multi-INTACT constructs two genetic instruments,

$$\begin{aligned} \hat{\boldsymbol{E}} &= \boldsymbol{G}\hat{\boldsymbol{\beta}}_e \\ \hat{\boldsymbol{P}} &= \boldsymbol{G}\hat{\boldsymbol{\beta}}_p, \end{aligned} \tag{5}$$

such that they are uncorrelated with $\boldsymbol{U}$ and $\boldsymbol{e}_Y$. The $p$-vectors $\hat{\boldsymbol{\beta}}_e$ and $\hat{\boldsymbol{\beta}}_p$ denote prediction weights for expression and protein levels derived from a TWAS method. Furthermore, as the colocalization prior effectively controls for the pleiotropic effects, we choose not to explicitly estimate $\boldsymbol{\beta}_Y$ and absorb its effect into the residual error $\boldsymbol{e}_Y$. In the end, we fit the following regression model to compute the marginal likelihood for $\gamma \neq 0$ or $\delta \neq 0$, i.e.,

$$Y = \gamma \hat{E} + \delta \hat{P} + e_Y. \tag{6}$$

Note that, the estimated percentage of variance explained (PVE), or $R^2$, by fitting (6) is identical to the squared canonical correlation between $Y$ and $(\hat{E}, \hat{P})$, which we reason from the perspective of multivariable IV analysis without the SEM formulation in Results section.

Because the regression model (6) requires only genetically predicted molecular gene products, the inference procedure can be naturally extended to multi-sample designs, in which the prediction models for $\hat{E}$ and $\hat{P}$ are learned in a cohort different from the GWAS samples. Importantly, the extension to data from multi-sample designs does not alter the causal implication of the original SEM model.

### Computing gene probability of putative causality

We compute the gene probability of putative causality for a target gene by applying the Bayes rule,

$$\text{GPPC} := \text{Pr}(\gamma \neq 0 \text{ or } \delta \neq 0 \mid \text{data}) \propto \pi f(p_{\text{coloc,E}}, p_{\text{coloc,P}}) \, \text{BF}, \tag{7}$$

where $p_{\text{coloc,E}}$ and $p_{\text{coloc,P}}$ denote the pre-computed gene-level colocalization probabilities of respective molecular QTLs and GWAS hits, $\pi f(p_{\text{coloc,E}}, p_{\text{coloc,P}})$ denotes the composite prior probability of putative causality, and BF represents the marginal likelihood/Bayes factor. See Supplemental Methods for details on the computation of BF, estimation of $\pi$, and the prior function $f$.

### Computing gene product relevance probability

The computation of gene product relevance probabilities breaks down to evaluating posterior probabilities for $M_E, M_P$, and $M_{P+E}$. To be consistent with the GPPC calculation, we specify $\text{Pr}(M_0) = 1 - \pi f(p_{\text{coloc,E}}, p_{\text{coloc,P}})$ and define the following conditional priors for the three alternative models,

$$\text{Pr}(M_E \mid \overline{M_0}) = h_E,$$
$$\text{Pr}(M_P \mid \overline{M_0}) = h_P,$$
$$\text{Pr}(M_{E+P}) \mid \overline{M_0}) = h_{E+P},$$

where $\overline{M_0}$ indicates the set of non-null models and $h_E + h_P + h_{E+P} = 1$.

Following an empirical Bayes procedure, we first compute the Bayes factor of each non-null model for all target genes and design an EM algorithm to obtain the MLE of $(h_E, h_P, h_{E+P})$ by pooling all candidate genes (Supplemental Methods). We then evaluate the required posterior model probabilities by plugging the estimated hyperparameters and subsequently compute the gene product relevance probabilities for each target gene using Eq. (1).

### Input for Multi-INTACT

Multi-INTACT is compatible with all existing TWAS methods that provide genotype prediction models for candidate gene products (e.g., TWAS Fusion, PTWAS, PrediXcan, and SMR) and probabilistic colocalization methods that quantify gene-level

colocalization evidence using probabilities (e.g., fastENLOC and coloc). In practice, we observe that inputs derived from TWAS and colocalization methods leveraging probabilistic multi-SNP fine-mapping results achieve the highest sensitivity for prioritizing PCGS among methods with proper FDR control [8]. Therefore, we strongly recommend performing multi-SNP fine-mapping analyses, using software packages DAP or SuSiE, for each gene product to generate optimal inputs for Multi-INTACT. We show an example of this pre-processing procedure in our real data analysis (Additional file 1: Supplemental Methods).

## Computation with GWAS summary statistics

The Multi-INTACT computation can be approximated using summary statistics of GWAS data. At a minimum, single-SNP association testing $z$ scores, weights for molecular phenotype prediction, and an appropriate LD reference panel are required to compute multivariate Wald statistic for Bayes factor calculation. We show the details of two approximate computation methods using the minimum GWAS summary statistics in Supplemental Methods. The same information can also be used to compute gene-level colocalization evidence.

It is worth emphasizing that summary statistics-based computation is not exact, and the loss of accuracy is expected. In practice, we find that if the LD reference panel matches well with the underlying GWAS samples, the results approximate the exact computation (using individual-level data) well (Additional file 1: Fig. S9). Most importantly, there should not be inflation of type I errors in PCG implication due to replacing individual-level data with the corresponding summary statistics under this setting. However, the consequence of severe mismatch between the LD panel and GWAS samples is unclear and needs further investigation.

## Simulation study

We use genotypes of 477K SNPs on chromosome 5 from 500 GTEx samples, including 1198 consecutive genes, each with at least 1500 common *cis*-SNPs. The complex and molecular phenotypes are simulated based on the complete DAG models shown in the second row of Additional file 1: Table S1, all of which assume no direct effects between *E* and *P* (note that Multi-INTACT inference does not assume or use this information).

Specifically, in each simulated data set, each of 1198 genes' causal mechanisms is independently drawn from a Multinomial($\pi_0, \pi_E, \pi_P, \pi_{E+P}$) distribution, where $\pi_0$, $\pi_E$, $\pi_P$, and $\pi_{E+P}$ represent probabilities of the null, expression-only, protein-only, and expression-and-protein models. For each gene, we randomly select two eQTLs and two pQTLs, where one variant is both a causal eQTL and pQTL. We select a distinct causal GWAS SNP. All effect sizes (represented by edges in the DAGs) are drawn from a $N(0, \phi^2)$ distribution, with $\phi$ set to 0.6 and residual error variances set to 1 to yield realistic signal-to-noise ratios. The distribution of the proportion of variance explained for all simulated phenotypes is shown in Additional file 1: Fig. S10. The Mean PVE for gene expression, protein levels, complex trait, expression-mediated complex trait, and protein-mediated complex trait are 0.167, 0.166, 0.159, 0.050, and 0.049, respectively. These values reasonably resemble the observed metabolite data in practice; the mean (SD) of the estimated

heritability across all metabolites is 0.186 (0.149). We use GCTA [75] to estimate metabolite heritability.

We simulate 100 datasets using the above scheme by varying the values of $(\pi_0, \pi_E, \pi_P, \pi_{E+P})$, with approximately 1/3 simulated datasets taking values from (0.8,0.1,0.05,0.05), (0.8,0.05,0.1,0.05), and (0.8,0.05,0.05,0.1), respectively. We use the different simulated datasets to examine the accuracy of the estimated $(h_E, h_P, h_{E+P})$ values from the proposed EM algorithm. The FDR and power for PCG discovery are calculated across all simulated datasets. Additionally, we vary the ratios of causal versus noncausal genes by subsampling the simulated data, the Multi-INTACT results remain stable (Additional file 1: Table S5).

In order to examine the robustness of Multi-INTACT in the presence of effects between expression and protein levels, we perform additional simulations to represent all 9 scenarios shown in Table S1. Additionally, we provide a comparison of Multi-INTACT to two additional PCG implication methods: SMR [11] and FOCUS [5], a causal gene method designed to control for LD hitchhiking. We describe the design of these additional simulations in Supplemental Methods. Power and FDR results for these simulations are shown in Additional file 1: Figs. S11–S13.

Finally, stratifying the analysis results of the simulated data reveals that the power of Multi-INTACT is positively correlated with the signal-to-noise ratios of genetic association signals (Additional file 1: Fig. S8). Thus, increasing sample sizes in molecular and complex trait association analyses is likely to enhance the power for PCG discovery.

### Preprocessing of UK Biobank pQTL data and multi-tissue GTEx eQTL data

We use PTWAS [15] multi-tissue prediction models trained on the GTEx data set to predict expression for the individuals in the METSIM cohort. For clarity, PTWAS refers to a method that can generate molecular phenotype prediction models, and it can be used to perform either TWAS or PWAS analysis if it is trained on expression or protein data, respectively. For PWAS analysis, we use the most significant cis-pQTL ($\pm 1$ Mb) to predict protein levels for each individual. We perform pairwise colocalization analyses between the QTL data and metabolite GWASs using fastENLOC, computing gene level colocalization probabilities [20] to quantify the likelihood of a colocalized variant for each metabolite-transcript or metabolite-protein pair.

### INTACT and Multi-INTACT analyses

We performed INTACT analyses using the default setting in the R package (linear prior and GLCP threshold $t = 0.05$). We focus on genes with both expression data and protein data available in the UK Biobank and GTEx prediction models. The number of tested genes per metabolite-tissue pair depends on the availability of expression and protein prediction models. The number of genes tested ranges from 186 (kidney cortex) to 1795 (nerve tibial). The complete breakdown is shown in Additional file 1: Fig. S14.

### INTACT-GSE pathway enrichment analysis for lipid metabolites

We perform probabilistic GSEA using the Multi-INTACT results derived from liver expression data. We use Multi-INTACT's default settings (truncation threshold equal to 0.05 with the linear prior function) to compute posterior probabilities for each

gene-metabolite triplet. For each tested gene, we compute an aggregated probability of putative causality across all lipid metabolites by

$$1 - \prod_i (1 - \mathrm{GPPC}_i),$$

where $\mathrm{GPPC}_i$ is the gene probability of putative causality for the examined gene and the $i$th lipid metabolite. Intuitively, the aggregated quantity represents the probability of the target gene is a PCG for at least one metabolite. We examine all GO BP terms for which at least one annotated gene has a nonzero aggregated probability of putative causality. GSEA input, including gene-level aggregated probabilities of putative causality, is included in the supplemental data. We access the GO term annotation data via the R package `org.Hs.eg.db` (v3.17.0).

## Supplementary information

Additional file 1: Figure S1: Various causal scenarios in which only one type of molecular QTL is colocalized with a causal GWAS hit. Figure S2: EM algorithm estimate distributions across simulated data sets. Figure S3-S5: Distributions of posteriors for each gene product-to-trait effect scenario and three posterior types. Figure S6: Multi-INTACT gene implication results, by tissue, with proportions implicated by marginal analyses. Figure S7: Overlap between genes implicated by TWAS in whole blood and PWAS. Figure S8: Multi-INTACT power stratified by genetic PVEs. Figure S9: Comparison of two multivariate regression test statistic approximations based on summary level-data to the statistic computed from individual-level data. Figure S10: Distributions of expression heritability, protein heritability, complex trait heritability per gene, expression-mediated complex trait heritability per gene, and protein-mediated complex trait heritability per gene in 100 simulated data sets. Figure S11-S13: Multi-INTACT power and realized FDR for simulations. Figure S14: Number of genes tested by Multi-INTACT across tissues. Table S1: Causal diagrams representing possible causal relationships between genotypes, molecular traits and, and a complex trait. Table S2: Simulation INTACT results across a variety of expression and protein prediction models. Table S3: Simulation Multi-INTACT results across a variety of expression and protein prediction models. Table S4: Simulation Multi-INTACT results using both individual-level and summary-level data. Table S5: Multi-INTACT results by varying causal vs. non-causal gene ratios in subsampled simulation data. Supplemental Methods [78–83].

Additional file 2: Peer review history.

### Authors' contributions
JO, FL, RPG, HKI, JM, CB, EBF, ML, MB, and XW conceived and designed the research. All authors performed the research. JO, XY, BR, JC, and XW analyzed the resulting data. JO, MB, and XW wrote the paper. All authors read and approved the final manuscript.

### Funding
This work is supported by grants NIH-R01-ES033634 (XW, RP, FL), NIH-R01-DK062370 (MB), NIH-R35-GM138121 (XW), NIH-R01-HL162574 (RP, FL), and NIH-R01-GM109215 (RP, FL).

### Data availability
The datasets generated and/or analyzed during the current study, as well as scripts necessary to reproduce analyses, are available on Github under AGPL-3.0 license at (https://github.com/jokamoto97/multi_intact_paper) and at Zenodo (https://doi.org/10.5281/zenodo.13955303) [76]. The Multi-INTACT software is accessible in the `INTACT` R package under AGPL-3.0 license on Github (https://github.com/jokamoto97/INTACT) and at Zenodo (https://doi.org/10.18129/B9.bioc.INTACT) [77].

## Declarations

### Review history
The review history is available as Additional file 2.

### Peer review information

### Ethics approval and consent to participate
Not applicable.

### Consent for publication
Not applicable.

**Competing interests**
EBF and JC are employees and stockholders of Pfizer. The remaining authors declare no competing interests.

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

## 