## [Additional file 2: Peer review history. · Genome Biology]

Review history

First round of review

Reviewer 1

Okamoto et al. proposed a novel statistical framework to jointly multiple gene products to identify causal genes for complex traits. I find this method very interesting and can be a great addition to the community. I listed my major and minor comments below:

Major comments

1. My primary concern is that authors compared Multi-INTACT with TWAS/PWAS and colocalization method. To me, Multi-INTACT/INTACT is a method that builds on existing TWAS/PWAS and colocalization to identify the putative causal genes, which looks like a "fine-mapping-like" method. It is not surprising that Multi-INTACT outperformed TWAS/PWAS and colocalization. In addition, FOCUS (Mancuso et al. Nature Genetics. 2019), as a fine-mapping method, essentially models the posterior distribution of TWAS Z scores accounting for the predicted expression correlations. Other fine-mapping methods such as GIFT (Liu et al. Nature Genetics. 2024) and FOGS (Wu and Pan Hum Genet. 2020) also either built on or leverage TWAS statistics to identify putative causal genes. It is more of a fair comparison for FOCUS, GIFT, and FOGS. In this way, authors can demonstrate that by leveraging multiple gene products (mRNA expression, protein expression, splicing etc.), Multi-INTACT can identify more accurate genes and pathways compared to single-gene product method (INTACT, PTWAS, FOCUS, GIFT, and FOGS). Therefore, if possible, it makes much more sense to compare Multi-INTACT to INTACT, PTWAS, FOCUS, GIFT, and FOGS for both simulations and real data unless authors have strong different opinions on this.

a. From Figure S9-S11, I saw authors used FOCUS and SMR methods. However, when I searched "FOCUS" or "SMR" in both method and results section, authors didn't mention these two at all, which is super confusing. In addition, "SMR" is similar to TWAS framework, which is not a fine-mapping method.

2. Line 91, the Multi-INTACT answers the question "do ANY of its gene products exert a potential causal effect on the trait of interest?". Suppose we use Multi-INTACT on mRNA and proteins, and the inference results give us the $P(\text{mRNA or Proteins})$, it is not surprising we see higher power comparing INTACT with only mRNA because $P(\text{mRNA or Proteins}) \geq P(\text{mRNA})$. I think it should be more correct to meta-analyze the INTACT (and other methods FOCUS, PTWAS, ..., etc) results across its single-gene versions. If results are z scores, then simple meta-analysis is good. If the results are PIPs, then $P_{\text{meta}} = 1 - (1 - P_{\text{mRNA}}) * (1 - P_{\text{proteins}})$ where P_{mRNA} and P_{proteins} are the PIP for single-gene results. I think this should also be applied to real data comparison.

3. My second primary concern is about the assumption of Multi-INTACT and how often can this method be practical in reality. Multi-INTACT essentially assumes that E and P have separate effects on Y (Figure 1 and Table S1 second row). However, based on the central dogma of the molecular biology, people may assume that protein expression can be further mediating product after mRNA expression (like table S1: third row, column 1 and 2). On the other hand, previous studies (Suhre et al. 2021. Nat Rev Genet; Suhre, K. et al. 2017 Nat Comm; Emilsson et al. Science. 2018; Sun et al. Nature. 2018) show only 40% of cis-pQTLs colocalized with cis-eQTLs, suggesting some post-transcriptional activities, which possibly aligns more to the assumption of Multi-INTACT. I would suggest authors to link their method assumption more to specific molecular mechanisms in the method overview section. That is, what kind of molecular mechanisms does Multi-INTACT capture (with references), and how can the central dogma of the molecular biology relate to

Multi-INTACT, can we see the DNA-RNA-Protein triad as a model misspecification for Multi-INTACT?

4. Figure 4, in the legend, it said TWAS, but the caption said its PTWAS.

5. I have following comments regarding the simulations:

a. Line 141, why did authors have 80%/20% ratio? Are there any reference? Can authors vary this parameter?

b. Figure S2, besides showing the boxplot, can authors show the mean and corresponding 95% confidence interval to see if it can cover the true proportion. If yes, that would be stronger evidence comparing to the way eyeballing if boxplot covers the true dotted line. If not, what are the possible reasons (power, algorithm-issue, or initial value for the hyperparameters)?

c. Figure 3, it would be great if authors can be more clear on the meaning of "posterior" for the y-axis.

d. Because Multi-INTACT can report gene product relevance probability, we are able to calculate the posterior probability of E->Y by summing over P(E) and P(E+P). For the same dataset, if I only run INTACT, and I can compute the power (I think this can be seen as a proxy of P(E->Y); similar for Proteins), do we see the similar power vs. P(E->Y) from Multi-INTACT? Do authors think this is a fair comparison?

e. It would be great if authors can show and explain the simulation performance when model assumption is not met in the result section. I only found that authors mentioned this from line 439 to line 442 in the method section, but they never talked about it in the results section.

f. Line 428, authors mentioned that all effect sizes are drawn from normal with variance ϕ^2 where ϕ is 0.6. As author mentioned, there are two cis-eQTLs, and the variance is 0.36; as a result, each single cis-eQTL explains 0.18 of expression variance (i.e., cis-SNP heritability), which is a relatively large number in practice (Wheeler et al. PLOS Genet. 2016; GTex paper; eQTLGen, etc.). Can authors provide references for this choice? And also varying this parameter and shows the results.

g. Line 432, can authors provide references for these numbers? The MESC paper stated that on average it's 0.11 explained by predicted expression for trait variance.

h. Authors also need to show how the sample size (eQTL, pQTL, and GWAS) changed the simulation results.

6. It would be great if authors can briefly mention their key findings through simulation and real data results along with take-home message/punchline for the readers.

7. Figure 1a, it would be great if authors can provide some basic explanation in the caption.

8. Line 203, "combining colocalization and PWAS evidence, identifies 293 PCGs", I think this relates to my primary concern that TWAS/PWAS only provides the marginal evidence for association testing. Because TWAS/PWAS statistics suffer from confounding of eQTL/pQTL effect sizes and predicted expression correlations (Wainberg et al. Nat Genet. 2019; Mancuso et al Nat Genet. 2019), I do not think we can call these TWAS significant genes "putative causal."

9. For the metabolite analysis, following the correlation analysis between TWAS and PWAS, what are the overlap proportion between TWAS significant genes and PWAS significant genes (Venn diagram).

10. Line 241-243, can authors quantify the significance here? Maybe some permutation test is needed. Authors showed that TWAS identified less genes and it is normal for it to identify less KBA genes; what it matters here is which method identifies gene sets that are more enriched in KBA genes. Again, following previous comments, I do not think TWAS/PWAS is an apple-to-apple comparison here.

11. When authors discuss the limitation of the study design and method, it would be great if authors can provide corresponding references showing some previous works that have made attempts to solve these or similar works.

12. In the discussion, it would be great if authors can discuss how a multi-ancestry study design can improve the results and how can it be used as one of the future directions (also with references). In addition, if analyzed data consists of admixed individuals, how can it affect the results? Authors don't need to show it in

simulations, some discussions are enough.

13. Color scheme for the methods is not consistent. For example, Multi-INTACT is blue color in figure 2 while Multi-INTACT is red color in Figure 4, and the red color in figure 3 means M_E(Expression). This can be very confusing to the readers.

Minor comments

1. Line 107, "exhibit modest colocalization", it would be great if authors can define/quantify "modest" here.
2. Line 303, it would be great if authors can be more clear on the definition of "distributional assumption".
3. Line 155-156, I think these are the first places that introduce comparing methods, it would be great if authors can provide corresponding references. For example, it takes me some time to find the original paper for INTACT.
4. At some places, "e.g." should be "e.g.,"
5. Mostafavi et al. 2023 Nature Genetics should be cited in the introduction when talking about the limitation of eQTLs in the GWAS settings. Wainberg et al. Nat Genet. 2019 should be also cited when talking about limitation of TWAS/PWAS.
6. Equation 2 of the method section, it would be great if authors can also define theta.
7. When introducing PTWAS, I think authors need to be clear and emphasize it's different from TWAS/PWAS. It can be confusing to readers who are not familiar with PTWAS; that is, they may think PTWAS is P/TWAS or TWAS/PWAS.

Reviewer 2

Thank you for the opportunity to review the paper entitled "Integrative analysis of the genome, transcriptome, and proteome identifies causal mechanisms of complex traits". This manuscript focuses on multi-integration of transcriptome-wide association studies and colocalization and proposes a new method called Multi-INTACT to implicate causal genes. By using simulations and real-data applications, they show Multi-INTACT has advantages over protein-alone or expression-alone integrative analyses. I believe this work will make an important contribution to the field but have questions and comments for the authors to consider to further improve the manuscript.

1. As described, Multi-INTACT is built based on instrumental variable inference procedure; a valid Multi-INTACT analysis must satisfy the three key model conditions including the relevance assumption, the independence assumption, the exclusion restriction assumption. Thus, it should be clearer to show how these assumptions are met by Multi-INTACT;
2. The output of the colocalization is one of the input of Multi-INTACT; however, it is not known which colocalization approaches are used in the Multi-INTACT method, and what the assumptions of colocalization are defined? One or multiple causal SNPs within a given gene are assumed? And if the former, what are the influence regarding the type I error and power on the performance of Multi-INTACT?
3. It is unclear how to calculate the local false discovery rate or false discovery rate based the posterior probability.
4. it is not known how to define the sample size of the phenotype (i.e., y) in the simulations? Is it also 500 as that of the expression and the protein? If that, then the simulations seem not to be in line with the real-data applications where the sample size of y generally is much larger than that of the expression and the protein.
5. Some minor errors include casual (should be casual) in figure 1 and in for (line 274) and will to (line 317).

Reviewer 3

- Are the methods appropriate to the aims of the study, are they well described, and are necessary controls included?

The authors extend single exposure IV analysis to account for multiple endogenous variables (e.g. expression, protein abundance, etc.), while integrating colocalization information as in their previous work INTACT, described in Okamoto et al. (2023). The goal is identification of a putative causal gene (PCG) for a downstream trait, while making use of individual or summary data for multiple molecular QTL.

- Are the conclusions adequately supported by the data shown?

The authors support their conclusions about Multi-INTACT through detailed simulation (causal gene identification through any gene product, and relevant gene product identification), and application to a set of eQTL and pQTL datasets (METSIM, UKB, and GTEx), where a subset of GWAS signals have been assigned to causal genes through manual data curation (Knowledge Based Analysis, KBA).

- Are sufficient details provided to allow replication and comparison with related analyses that may have been performed?

There are sufficient details.

It would be helpful to compare some of the results in the context of recent related analyses. A recent preprint by Tambets et al. (10.1101/2023.09.29.560109) also investigates concordance of effect sizes across eQTL and pQTL following colocalization, where precision of 70-80% is found, where precision refers to expected direction of effect of IVs on gene expression and on protein abundance where the protein is matched to the gene.

- Is the method likely to be of broad utility? Is any software component easy to install and use? Please indicate briefly the novel features and/or advantages of the method, and/or please reference the relevant publications and which methods, if any, it should be compared with.

The software is easily installed and the methods are provided on Bioconductor and GitHub. The software vignette includes an example of running Multi-INTACT. Some more explanation for how non-statisticians can interpret the output would be helpful.

The vignette currently has:

"To compute gene probabilities of putative causality (GPPCs) and gene product relevance probabilities (GPRPs), run: ..."

"The output from the multi_intact function is a list object containing 3 items. The first is a data frame with the GPPC, GPRP for expression (GPRP_1), and for protein (GPRP_2). The second is a numeric 3-vector containing conditional prior parameter estimates The third is a Boolean indicating whether the EM algorithm converged."

It would be helpful to further interpret the output for a more biological audience. Maybe an example of how the results could be visualized.

Additionally, the vignette uses pre-prepared datasets but doesn't show where such values would come from. It mentions "These are provided as output by most popular TWAS and colocalization methods...If the user wishes to specify TWAS Bayes factors instead of z-scores, they can do so through the argument `twas_BFs...`" It might be helpful to explicit list in the vignette which upstream tools could or should be used here, or what methods the authors have used or recommend upstream of Multi-INTACT.

- Is the paper of broad interest to others in the field, or of outstanding interest to a broad audience of biologists?

The paper in theory is of broad interest to others in the field, although there is some language in the main text that isn't fully explained and would not be easily understood by a broad audience. For example "do-calculus" part starting on line 88, and then the brief mention of the prior formulation, "the prior incorporates the colocalization evidence of molecular QTLs and GWAS hits..." For the prior formulation, it is worth describing more detail in the main text about how this happens in Multi-INTACT, as it seems a key point about the method and its benefit over methods that don't incorporate colocalization evidence into PCG inference.

Additional comments:

1. In Figure 1 the distinction between solid and dashed edges should be described.
2. Does the statistical power for the eQTL and pQTL studies affect the ability to distinguish between E->P and P->E?
3. What is the percent variance explained (population-level PVE) for the simulation studies? How does this compare to the other datasets shown in the Results?
4. In Figure 3 and the supplementary figures showing "distributions of posteriors for each gene product-to-trait effect scenario", is there something that characterizes simulated loci when the correct posterior type has low posterior probability? Dense LD? Any intuition about what characteristics of loci may lead to lower sensitivity?
5. For the 30% of annotated KBA pairs that are not found by Multi-INTACT, can the authors detect any patterns (maybe unmodeled mechanisms) that would lead to further method development and higher power? Also for the analysis of the concordance of direction of gene product through expression and protein abundance, are there other mechanisms that might lead to appearance of mis-matching direction?
6. What background set is used in the gene set analysis (lines 281 onward)?

Authors' response to reviewers

We sincerely appreciate the valuable and insightful feedback from all three reviewers. We provide a point-by-point response to each comment in below. The reviewers' comments are in black, and our responses are highlighted in blue. We have highlighted the corresponding changes in the main text and supplemental material in red font.

Reviewer 1

Okamoto et al. proposed a novel statistical framework to jointly multiple gene products to identify causal genes for complex traits. I find this method very interesting and can be a great addition to the community. I listed my major and minor comments below:

Major comments

1. My primary concern is that authors compared Multi-INTACT with TWAS/PWAS and colocalization method. To me, Multi-INTACT/INTACT is a method that builds on existing TWAS/PWAS and colocalization to identify the putative causal genes, which looks like a “fine-mapping-like” method. It is not surprising that Multi-INTACT outperformed TWAS/PWAS and colocalization. In addition, FOCUS (Mancuso et al. Nature Genetics. 2019), as a fine-mapping method, essentially models the posterior distribution of TWAS Z scores accounting for the predicted expression correlations. Other fine-mapping methods such as GIFT (Liu et al.

Nature Genetics. 2024) and FOGS (Wu and Pan Hum Genet. 2020) also either built on or leverage TWAS statistics to identify putative causal genes. It is more of a fair comparison for FOCUS, GIFT, and FOGS. In this way, authors can demonstrate that by leveraging multiple gene products (mRNA expression, protein expression, splicing etc.), Multi-INTACT can identify more accurate genes and pathways compared to single-gene product method (INTACT, PTWAS, FOCUS, GIFT, and FOGS). Therefore, if possible, it makes much more sense to compare Multi-INTACT to INTACT, PTWAS, FOCUS, GIFT, and FOGS for both simulations and real data unless authors have strong different opinions on this.

We thank the reviewer for this comment. The fine-mapping methods mentioned, such as FOCUS, GIFT, and FOGS, simultaneously evaluate multiple genes within a genomic region. In contrast, our proposed multi-INTACT approach analyzes one gene at a time, similar to methods like INTACT, PTWAS, and colocalization analyses. Therefore, we do not consider Multi-INTACT to be a fine-mapping approach. We agree with the reviewer that both multi-gene fine-mapping methods and multi-INTACT are likely to outperform traditional single-gene TWAS and colocalization methods. However, the sources of their power gains differ, which complicates direct comparisons between these methodologies. Nonetheless, we are indeed interested in extending Multi-INTACT to enable simultaneous analysis of multiple genes in our future work.

a. From Figure S9-S11, I saw authors used FOCUS and SMR methods. However, when I searched “FOCUS” or “SMR” in both method and results section, authors didn’t mention these two at all, which is super confusing. In addition, “SMR” is similar to TWAS framework, which is not a fine-mapping method.

We have added the references to FOCUS and SMR to the last paragraph of Section 5.5.

2. Line 91, the Multi-INTACT answers the question “do ANY of its gene products exert a potential causal effect on the trait of interest?”. Suppose we use Multi-INTACT on mRNA and proteins, and the inference results give us the $P(\text{mRNA or Proteins})$, it is not surprising

we see higher power comparing INTACT with only mRNA because $P(\text{mRNA or Proteins}) \geq P(\text{mRNA})$. I think it should be more correct to meta-analyze the INTACT (and other methods FOCUS, PTWAS, ..., etc) results across its single-gene versions. If results are z scores, then simple meta-analysis is good. If the results are PIPs, then $P_{meta} = 1 - (1 - P_{mRNA}) * (1 - P_{proteins})$ where P_{mRNA} and $P_{proteins}$ are the PIP for single-gene results. I think this should also be applied to real data comparison.

We agree with the reviewers reasoning on the power increase by considering multiple gene products simultaneously. However, the reviewers suggestion to combine PIPs relies on the assumption that the marginal mRNA-to-trait and protein-to-trait effects are independent. In practice, this assumption may not hold, particularly in cases where mRNA-to-trait effects are partially or fully mediated by proteins (an example of this kind is given in the next response to comment 3). To avoid this strong assumption, Multi-INTACT explicitly models and computes $P(\text{mRNA or Protein})$ via the more general inclusion-exclusion principle. Therefore, we view the reviewers proposal as a special case of the broader Multi-INTACT framework. A similar argument applies to z-scores, as meta-analysis methods also require independent or conditionally independent data structures.

3. My second primary concern is about the assumption of Multi-INTACT and how often can this method be practical in reality. Multi-INTACT essentially assumes that E and P have separate effects on Y (Figure 1 and Table S1 second row). However, based on the central dogma of the molecular biology, people may assume that protein expression can be further mediating product after mRNA expression (like table S1: third row, column 1 and 2). On the other hand, previous studies (Suhre et al. 2021. Nat Rev Genet; Suhre, K. et al. 2017 Nat Comm; Emilsson et al. Science. 2018; Sun et al. Nature. 2018) show only 40% of cis-pQTLs colocalized with cis-eQTLs, suggesting some post-transcriptional activities, which possibly aligns more to the assumption of Multi-INTACT. I would suggest authors to link their method assumption more to specific molecular mechanisms in the method overview section. That is, what kind of molecular

mechanisms does Multi-INTACT capture (with references), and how can the central dogma of the molecular biology relate to Multi-INTACT, can we see the DNA-RNA-Protein triad as a model misspecification for Multi-INTACT?

We thank the reviewer for the thoughtful comment. In response, we have revised Section 5.1 of the manuscript to clarify that our model does not assume E and P necessarily have independent effects on Y . Specifically, we now highlight that the Multi-INTACT model, as represented by the structural equations, is a *conditional model* (represented by the do-calculus on page 6) that focuses on the potential causal relationships between $E \rightarrow Y$ and $P \rightarrow Y$, without requiring a full specification of other possible causal relationships (e.g., between E and P). As a result, the inferred $E \rightarrow Y$ and $P \rightarrow Y$ effects do not necessarily correspond to the marginal causal effects, but rather to the “net effects” after accounting for the relationship between E and P . To illustrate, consider a causal path $E \rightarrow P \rightarrow Y$, where E has a marginal effect on Y that is fully mediated by P . In this scenario, Multi-INTACT would detect that the $E \rightarrow Y$ effect is zero because $\hat{E} \perp\!\!\!\perp Y \mid \hat{P}$ from the observed data without making explicit causal assumptions between E and P . More generally, we view the conditional model by Multi-INTACT is compatible with all the graphical models presented in Table S1.

4. Figure 4, in the legend, it said TWAS, but the caption said its PTWAS. We

have changed the caption to say TWAS to match the legend.

5. I have following comments regarding the simulations:

a. Line 141, why did authors have 80%/20% ratio? Are there any reference? Can authors vary this parameter?

This is an excellent question, and we appreciate the reviewer’s suggestion. In our simulation scheme, we randomly assign causal gene status to each candidate gene using a Bernoulli distribution with a probability of 0.2. This resulted in 24,436 out of 119,700 genes, or approximately

20%, being designated as causal across 100 simulated datasets. This approach ensures a balanced representation of causal and non-causal genes in the simulated data, allowing for accurate benchmarking of false discovery rates and power for our proposed method and its comparisons. Importantly, the ratio of causal to non-causal genes is not used in our analysis of the simulated data. Following the reviewers suggestion, we subsample the data to create varying ratios of non-causal to causal genes. The results, now summarized in Supplemental Table S5, demonstrate that false discovery rates were consistently controlled at the 5% level across all examined ratios, and power estimates remained stable across different ratios of causal to non-causal genes.

In practice, the true ratio of causal to non-causal genes is unknown. Existing theories, such as the polygenic and omnigenic models, suggest that a significant proportion, potentially the majority, of genes could be causal for any trait. Our previous study on metabolite causal genes (Yin et al., 2022) identifies nearly 4,000 unique causal genes across metabolites using TWAS analysis, representing approximately 20% of protein-coding genes. However, the number of experimentally validated causal genes remains relatively low.

b. Figure S2, besides showing the boxplot, can authors show the mean and corresponding 95% confidence interval to see if it can cover the true proportion. If yes, that would be stronger evidence comparing to the way eyeballing if boxplot covers the true dotted line. If not, what are the possible reasons (power, algorithm-issue, or initial value for the hyperparameters)?

We have added another row of plots to Figure S2 to show the mean and 95% confidence interval. The 95% confidence intervals all cover the true proportion.

c. Figure 3, it would be great if authors can be more clear on the meaning of "posterior" for the y-axis.

We thank the reviewer for the suggestion. We have added to the caption of Figure 3 to clarify the meaning of "posterior" for the y-axis.

d. Because Multi-INTACT can report gene product relevance probability, we are able to calculate the posterior probability of $E \rightarrow Y$ by summing over $P(E)$ and $P(E+P)$. For the same dataset, if I only run INTACT, and I can compute the power (I think this can be seen as a proxy of $P(E \rightarrow Y)$; similar for Proteins), do we see the similar power vs. $P(E \rightarrow Y)$ from Multi-INTACT? Do authors think this is a fair comparison?

The left panel of Figure 4 includes the comparison of the performance of Multi-INTACT expression GPRP with the INTACT posterior probability in detecting an $E \rightarrow Y$ effect. It is evident that Multi-INTACT outperforms both INTACT and methods that analyze a single molecular trait at a time. When E is the only molecular phenotype with a causal effect on Y , Multi-INTACT and marginal INTACT analyses are expected to have comparable power. However, if P also has a direct causal effect on Y , controlling for $P \rightarrow Y$ while estimating the effect of $E \rightarrow Y$ enhances the signal-to-noise ratio. This is analogous to how multi-SNP fine-mapping is more powerful than single-SNP analysis in genetic association studies. A similar pattern is observed when comparing the Multi-INTACT protein GPRP with the INTACT posterior in detecting $P \rightarrow Y$ effects, as shown in the right panel of Figure 4.

e. It would be great if authors can show and explain the simulation performance when model assumption is not met in the result section. I only found that authors mentioned this from line 439 to line 442 in the method section, but they never talked about it in the results section.

The text between lines 439 and 442 explains the following point: The simulation data are derived from a comprehensive set of directed acyclic graphs (DAGs), which fully define the causal relationships between genetic variants (G), molecular phenotypes (E and P), and the complex trait Y , as outlined in the Methods section and supplementary materials. As previously discussed, the Multi-INTACT method does not rely on or assume specific causal relationships between G , E , and P . Instead, it focuses on assessing potential causal effects from $E \rightarrow Y$ and $P \rightarrow Y$. In conclusion, our analysis model differs from the data-generating models used in the simulations, and

the results demonstrate that Multi-INTACT is robust across a range of probabilistic generative models.

f. Line 428, authors mentioned that all effect sizes are drawn from normal with variance ϕh^2 where ϕ is 0.6. As author mentioned, there are two cis-eQTLs, and the variance is 0.36; as a result, each single cis-eQTL explains 0.18 of expression variance (i.e., cis-SNP heritability), which is a relatively large number in practice (Wheeler et al. PLOS Genet. 2016; GTeX paper; eQTLGen, etc.). Can authors provide references for this choice? And also varying this parameter and shows the results.

We understand the reviewers concern. Since we neither simulate data from a linear mixed model nor normalize the real genotypes used in the genetic association analysis, the reviewers calculation does not fully reflect the characteristics of our simulated data. In Supplementary Figure S8, we show the distribution of the realized PVE by cis-eQTLs across all simulated genes, where PVE (R^2) is calculated by regressing the simulated gene expressions on the genotypes of all the true eQTLs for each gene.

Although the mean PVE (With all cis-eQTLs) per gene is approximately 0.167, the distribution is significantly right-skewed, indicating an excessive small PVE values. Additionally, Wheeler et al.s (2016) analysis of DGN blood eQTL data yields a mean h^2 estimate of 0.149 (Table 1, row 1), which closely matches our simulation settings.

g. Line 432, can authors provide references for these numbers? The MESC paper stated that on average it's 0.11 explained by predicted expression for trait variance.

These numbers are derived from the histograms of the different types of realized PVEs shown in Supplementary Figure S8. To clarify, we do not use these numbers to generate simulated data; rather, they serve as summary statistics obtained from the simulated data under the specified data-generating scheme. Notably, the PVEs based on predicted expression levels show substantial variability across simulated complex traits, aligning with the findings of the MESC paper. In

our simulations, the average PVE by predicted gene expression (0.05) is slightly lower than the value reported in the MESC paper, though we consider them comparable. Nevertheless, we acknowledge that our simulation setting may be more challenging than the practical setting, which could lead to a potential underestimation of the Multi-INTACT power in practice.

h. Authors also need to show how the sample size (eQTL, pQTL, and GWAS) changed the simulation results.

We understand the reviewers point. The sample size is a critical factor influencing the genetic signal-to-noise ratio in identifying causal genetic associations and PCGs. However, since our simulations are based on real genotype data, adjusting the sample size while maintaining realistic LD patterns presents a challenge. In response to the reviewers suggestion, we propose that the effect of varying sample sizes can be demonstrated by stratifying the realized genetic PVEs for expression, protein, and GWAS data, which directly measure genetic signal-to-noise ratios. We have added a supplementary figure (Figure S14) showing that higher signal-to-noise ratios in genetic association data enhance the power to detect PCGs. Additionally, we have included the following statement in the Results section:

Finally, stratifying the analysis results of the simulated data reveals that the power of Multi-INTACT is positively correlated with the signal-to-noise ratios of genetic association signals (Figure S14). Thus, increasing sample sizes in molecular and complex trait association analyses is likely to enhance the power for PCG discovery.

6. It would be great if authors can briefly mention their key findings through simulation and real data results along with take-home message/punchline for the readers.

We thank the reviewer for this comment. We have summarized our take-home message in the first paragraph of the Discussion section:

“We present Multi-INTACT, a novel statistical method for mechanism-aware PCG implication

by integrating GWAS and multiple types of molecular QTL data. Through simulations and real data analysis, we demonstrate that Multi-INTACT properly controls type I errors by effectively guarding against LD hitchhiking and identifies causal genes with increased statistical power compared to other popular PCG implication methods. Additionally, its ability to simultaneously evaluate the relevance of multiple gene products (i.e., those exerting direct effects on complex traits) can help gain insights on detailed molecular mechanisms of complex diseases, making it a promising tool for drug target discovery.”

7. Figure 1a, it would be great if authors can provide some basic explanation in the caption.

We have updated the caption of Figure 1 to provide a basic explanation for Figure 1a.

8. Line 203, “combining colocalization and PWAS evidence, identifies 293 PCGs”, I think this relates to my primary concern that TWAS/PWAS only provides the marginal evidence for association testing. Because TWAS/PWAS statistics suffer from confounding of eQTL/pQTL effect sizes and predicted expression correlations (Wainberg et al. Nat Genet. 2019; Mancuso et al Nat Genet. 2019), I do not think we can call these TWAS significant genes “putative causal.”

We fully agree with the reviewers point, which is a key motivation behind developing the INTACT and Multi-INTACT methods. Many researchers[1, 2, 3, 4] have highlighted connections between TWAS, Mendelian randomization, and instrumental variable (IV) analysis, with the latter two being formal causal inference methods. Conceptually, if the causal assumptions of IV analysis are fully met, TWAS associations could establish a causal link between genes and complex traits. However, in practice, phenomena like widespread LD hitchhiking often violate these assumptions. Our method aims to detect such violations and enforce the necessary causal assumptions, as demonstrated in both simulations and real data analyses.

9. For the metabolite analysis, following the correlation analysis between TWAS and PWAS, what are the overlap proportion between TWAS significant genes and PWAS significant genes (Venn diagram).

We thank the reviewer for the suggestion. We have added a Venn diagram to compare the PWAS findings with the TWAS findings from the GTEx whole blood (Figure S13).

10. Line 241-243, can authors quantify the significance here? Maybe some permutation test is needed. Authors showed that TWAS identified less genes and it is normal for it to identify less KBA genes; what it matters here is which method identifies gene sets that are more enriched in KBA genes. Again, following previous comments, I do not think TWAS/PWAS is an apple-to-apple comparison here.

We understand the reviewer's point. Our comparison emphasizes the number of discoveries validated by the KBA, rather than the enrichment. It's important to note that while the KBA genes reflect the state-of-the-art in current knowledge, the set may still be incomplete. Crucially, a gene not included in the current KBA set could still be a genuine causal gene (e.g., the *GSTAI* example described in the manuscript). From this standpoint, enrichment testing, which relies on an imperfect gold standard and forces a partitioning of all genes into causal and non-causal categories, can be problematic. This issue has been demonstrated and discussed in a recent paper [5].

11. When authors discuss the limitation of the study design and method, it would be great if authors can provide corresponding references showing some previous works that have made attempts to solve these or similar works.

We have added references to the discussion section to describe how previous studies have approached some limitations of Multi-INTACT. We note that these other methods may be complementary to Multi-INTACT in integrative analysis.

12. In the discussion, it would be great if authors can discuss how a multi-ancestry study design can improve the results and how can it be used as one of the future directions (also with references). In addition, if analyzed data consists of admixed individuals, how can it affect the results? Authors don't need to show it in simulations, some discussions are enough.

We thank the reviewer for the suggestion. We have added a discussion of how a multi-ancestry study design can improve the results to the final paragraph of Section 3.

13. Color scheme for the methods is not consistent. For example, Multi-INTACT is blue color in figure 2 while Multi-INTACT is red color in Figure 4, and the red color in figure 3 means $M_E(\text{Expression})$. This can be very confusing to the readers.

We have updated Figures 3 and 4 so that the color scheme matches as much as possible. In Figures 1 and 3, blue represents expression analysis, red represents protein analysis, and purple represents a joint expression and protein analysis. We have updated Figure 4 so that the Multi-INTACT line is purple, representing the joint analysis. The color scheme in Figure 2 is unrelated to the type of method or molecular phenotype, so we left it as is.

Minor comments

1. Line 107, "exhibit modest colocalization", it would be great if authors can define/quantify "modest" here.

We have added the phrase i.e., has a gene-level colocalization probability of greater than zero, to clarify modest.

2. Line 303, it would be great if authors can be more clear on the definition of "distributional assumption".

We have added "Specifically, the graphical model does not assume normality of each phenotype or additivity (see the SEM in Methods)," directly after the sentence that is referenced by this comment.

3. Line 155-156, I think these are the first places that introduce comparing methods, it would be great if authors can provide corresponding references. For example, it takes me some time to find the original paper for INTACT.

We have added corresponding references for the colocalization, TWAS, and INTACT analyses used in the comparison of methods to the second paragraph of section 2.2.

4. At some places, "e.g." should be "e.g.,"

We have updated this error in the abstract.

5. Mostafavi et al. 2023 Nature Genetics should be cited in the introduction when talking about the limitation of eQTLs in the GWAS settings. Wainberg et al. Nat Genet. 2019 should be also cited when talking about limitation of TWAS/PWAS.

We have cited Mostafavi et al. 2023 Nature Genetics in the first sentence of the third paragraph of the introduction. We have cited Wainberg et al. 2019 Nature Genetics in the last sentence of the second paragraph of the introduction.

6. Equation 2 of the method section, it would be great if authors can also define theta.

We have added the sentence "Scalars θ_E and θ_P denote effects of unobserved confounding on expression and protein levels, respectively," to the first paragraph of section 5.1.

7. When introducing PTWAS, I think authors need to be clear and emphasize it's different from TWAS/PWAS. It can be confusing to readers who are not familiar with PTWAS; that is, they may think PTWAS is P/TWAS or TWAS/PWAS.

We added the sentence "For clarity, PTWAS refers to a method that can generate molecular phenotype prediction models, and it can be used to perform either TWAS or PWAS analysis if it is trained on expression or protein data, respectively," to section 5.6 when introducing PTWAS.

Reviewer 2

Thank you for the opportunity to review the paper entitled "Integrative analysis of the genome, transcriptome, and proteome identifies causal mechanisms of complex traits". This manuscript focuses on multi-integration of transcriptome-wide association studies and colocalization and proposes a new method called Multi-INTACT to implicate causal genes. By using simulations and real-data applications, they show Multi-INTACT has advantages over protein-alone or expression-alone integrative analyses. I believe this work will make an important contribution to the field but have questions and comments for the authors to consider to further improve the manuscript.

1. As described, Multi-INTACT is built based on instrumental variable inference procedure; a valid Multi-INTACT analysis must satisfy the three key model conditions including the relevance assumption, the independence assumption, the exclusion restriction assumption. Thus, it should be clearer to show how these assumptions are met by Multi-INTACT;

We appreciate the reviewer's comment. To the best of our knowledge, validating causal assumptions—particularly the IV analysis independence/randomization and exclusion restriction assumptions—remains an open problem in causal inference based on observational data [6]. In practice, sensitivity analysis methods are often used to detect severe violations of these IV assumptions [6, 7]. This strategy is adopted by TWAS methods such as PTWAS [4], which we use as the default TWAS method in both our simulation and real data analyses.

We view Multi-INTACT as an improvement over standard TWAS analysis, providing a more robust framework for addressing the three key IV assumptions. The relevance assumption requires that the genetic instruments are strongly associated with the exposure (i.e., the molecular phenotypes). TWAS methods meet this criterion by explicitly selecting strong eQTLs/pQTLs to build molecular phenotype prediction models. Since Multi-INTACT directly uses these TWAS

molecular phenotype prediction models, we argue that it satisfies the relevance assumption in the same way as TWAS. To clarify this point, we have added the sentence: “The p -vectors $\hat{\beta}_E$ and $\hat{\beta}_p$ denote prediction weights for expression and protein levels derived from a TWAS method under Equation (5) in Methods.

Mendelian randomization relies on Mendel’s law of independent assortment to argue that genetic variants are valid instruments, thereby satisfying the independence assumption. Multi-INTACT uses the canonical correlation between complex trait levels and genetically-predicted molecular phenotypes as input, which can be viewed as a robust multivariable Mendelian randomization analysis (see Equation 6 and the accompanying paragraph in Methods). Thus, we argue that Multi-INTACT satisfies the independence assumption in the same way as Mendelian randomization.

The exclusion restriction cannot be empirically tested [8, 9], meaning it is impossible to prove definitively that the assumption holds. Therefore, we do not claim that Multi-INTACT analysis always satisfies this assumption. However, the key idea behind Multi-INTACT is that it is more robust to violations of the exclusion restriction caused by linkage disequilibrium [6], which often leads to excessive false positives in TWAS analyses [10, 11, 12]. In the third paragraph of section 2.1, we discuss in detail how we incorporate colocalization evidence to mitigate potential violations of the exclusion restriction.

2. The output of the colocalization is one of the input of Multi-INTACT; however, it is not known which colocalization approaches are used in the Multi-INTACT method, and what the assumptions of colocalization are defined? One or multiple causal SNPs within a given gene are assumed? And if the former, what are the influence regarding the type I error and power on the performance of Multi-INTACT?

We use the colocalization method implemented in fastENLOC to generate the colocalization input for Multi-INTACT (Section 5.6). Multi-INTACT is compatible with various probabilistic

colocalization methods, including fastENLOC, coloc, and eCAVIAR. We recommend fastENLOC because it estimates informative colocalization priors by performing enrichment analysis and, importantly, does not assume a single causal variant per gene. As described by Wen et al. (2017) [13], enrichment estimation is critical in colocalization analysis, as it allows the user to avoid strong assumptions about the colocalization priors. Our previous work, Hukku et al. (2021) [14], demonstrates that limiting the analysis to at most one causal variant per cis-region can significantly reduce the power of colocalization analysis (while the type I errors are not inflated).

3. It is unclear how to calculate the local false discovery rate or false discovery rate based the posterior probability.

We have added relevant references to the local fdr and FDR control procedures based on posterior probabilities, including [15, 16, 17, 18, 19]. A brief summary of this topic can be found at the end of Section 2.3 in Stephens (2017) [20].

4. it is not known how to define the sample size of the phenotype (i.e., y) in the simulations? Is it also 500 as that of the expression and the protein? If that, then the simulations seem not to be in line with the real-data applications where the sample size of y generally is much larger than that of the expression and the protein.

We understand the reviewer's point. Sample size directly affects the signal-to-noise ratio of genuine genetic associations and plays a crucial role in our subsequent analysis of PCGs. In our simulation studies using individual-level data, we employ real genotype data to capture realistic LD patterns. While our sample sizes do not match those of large GWAS studies, we carefully select genetic effect size simulation parameters to ensure that the realized heritability/PVE, which directly measures genetic signal-to-noise ratio, from the simulated complex trait data aligns with the observed GWAS data (see Supplementary Figure S8). Additionally, please refer to our response to Reviewer 1's comment 5.h.

5. Some minor errors include casual (should be casual) in figure 1 and in for (line 274) and will to (line 317).

Thank you for catching these errors. We have fixed all of them.

Reviewer 3

- Are the methods appropriate to the aims of the study, are they well described, and are necessary controls included?

The authors extend single exposure IV analysis to account for multiple endogenous variables (e.g. expression, protein abundance, etc.), while integrating colocalization information as in their previous work INTACT, described in Okamoto et al. (2023). The goal is identification of a putative causal gene (PCG) for a downstream trait, while making use of individual or summary data for multiple molecular QTL.

- Are the conclusions adequately supported by the data shown?

The authors support their conclusions about Multi-INTACT through detailed simulation (causal gene identification through any gene product, and relevant gene product identification), and application to a set of eQTL and pQTL datasets (METSIM, UKB, and GTEx), where a subset of GWAS signals have been assigned to causal genes through manual data curation (Knowledge Based Analysis, KBA).

- Are sufficient details provided to allow replication and comparison with related analyses that may have been performed?

There are sufficient details.

It would be helpful to compare some of the results in the context of recent related analyses. A recent preprint by Tambets et al. (10.1101/2023.09.29.560109) also investigates concordance of effect sizes across eQTL and pQTL following colocalization, where precision of 70-80% is found, where precision refers to expected direction of effect of IVs on gene expression and on protein abundance where the protein is matched to the gene.

We appreciate the reviewer's comment. In our analysis, we examine the directional concordance

between expression-to-trait and protein-to-trait effects for genes implicated by Multi-INTACT. In contrast, Tambets et al. focus on the directional concordance of genetic effects from variant-to-expression and variant-to-protein, conditioned on colocalization evidence. Although both approaches address important questions, they target different scientific problems and require distinct analytical strategies. As such, Multi-INTACT is not suitable for the type of analysis posed by Tambets et al., and direct comparisons between our results and theirs may not be straightforward.

Nevertheless, we have compared our findings to a similar analysis from recent literature. In Section 2.3, we refer to Zhang et al. (2023, 10.1101/2023.08.17.553749), who estimate the correlation between signed negative log p-values from TWAS and PWAS for four lipid traits to be in the range of 0.083 to 0.144. We replicated this analysis across all gene-metabolite-tissue pairs in our dataset and found a mean correlation of 0.156, which is consistent with the values reported by Zhang et al. However, when restricting the analysis to genes implicated by Multi-INTACT, the mean correlation approximately doubled to 0.319. Altogether, these results may suggest that Multi-INTACT has higher specificity, and its implicated PCGs exhibit a higher degree of directional concordance in gene product-to-trait effects.

- Is the method likely to be of broad utility? Is any software component easy to install and use? Please indicate briefly the novel features and/or advantages of the method, and/or please reference the relevant publications and which methods, if any, it should be compared with.

The software is easily installed and the methods are provided on Bioconductor and GitHub. The software vignette includes an example of running Multi-INTACT. Some more explanation for how non-statisticians can interpret the output would be helpful.

The vignette currently has:

”To compute gene probabilities of putative causality (GPPCs) and gene product relevance probabilities (GPRPs), run: ...”

”The output from the multi-intact function is a list object containing 3 items. The first is a data frame with the GPPC, GPRP for expression ($GPRP_1$), and for protein ($GPRP_2$). The second is a numeric 3-vector containing conditional prior parameter estimates The third is a Boolean indicating whether the EM algorithm converged.”

It would be helpful to further interpret the output for a more biological audience. Maybe an example of how the results could be visualized.

We have added the following chunk to the software vignette section 10 in order to aid non-statisticians in interpreting the output:

“The GPPC is a form of probabilistic evidence that a target gene exerts a causal effect on the complex trait through at least one of the provided gene product data types. The GPRPs provide probabilistic evidence to determine the gene product(s) that exert a direct effect on the complex trait. For example, in the output above, based on the Multi-INTACT output, it is very likely that the gene ENSG0000038274 has a causal effect, and it is likely that encoded protein levels exert a direct effect. There is relatively little evidence that the genes expression levels exert a direct effect (although it remains possible that gene expression exerts an effect that is mediated by protein levels). The prior parameter estimates represent prior probabilistic evidence of three possible underlying causal models (in which only expression, only protein levels, or both expression and protein levels exert an effect, respectively). Estimation of these prior parameters is required to estimate GPRPs.”

We also added three histograms to show the distribution of Multi-INTACTs probabilistic output. We included a discussion of the visualizations as well:

“From the histograms above, we can see that the probabilistic evidence of a causal effect on the complex trait is low for most genes (top plot). In the bottom plot, we display the distribution of GPRPs. Recall that all genes have 2 GPRPs ($GPRP_1$ for protein evidence, and $GPRP_2$ for expression), so we visualize the distribution of each GPRP type in each panel. Additionally, we

can see that the distribution of probabilistic evidence of a direct protein effect is similar to that of probabilistic evidence of a direct expression effect. These observed qualities are by design, as this is simulated data.”

Additionally, the vignette uses pre-prepared datasets but doesn't show where such values would come from. It mentions “These are provided as output by most popular TWAS and colocalization methods...If the user wishes to specify TWAS Bayes factors instead of z-scores, they can do so through the argument *twasBFs*...” It might be helpful to explicit list in the vignette which upstream tools could or should be used here, or what methods the authors have used or recommend upstream of Multi-INTACT.

We have added the following chunk to section 5 of the vignette:

“Some TWAS methods that we utilize in our work are PTWAS, PrediXcan, and SMR. We recommend using fastENLOC for colocalization analysis, as it estimates enrichment of QTL among GWAS hits and does not require specification of prior probabilities.”

We have included links within the chunk to the software repositories for each referenced method.

- Is the paper of broad interest to others in the field, or of outstanding interest to a broad audience of biologists?

The paper in theory is of broad interest to others in the field, although there is some language in the main text that isn't fully explained and would not be easily understood by a broad audience. For example “do-calculus” part starting on line 88, and then the brief mention of the prior formulation, “the prior incorporates the colocalization evidence of molecular QTLs and GWAS hits...” For the prior formulation, it is worth describing more detail in the main text about how this happens in Multi-INTACT, as it seems a key point about the method and its benefit over methods that don't incorporate colocalization evidence into PCG inference.

We thank the reviewer for this comment. We have edited the section mentioning “do-calculus” (2.1) to specify that we are using language from the field of causal models, and we have cited Judea Pearl’s book on causal models as a reference:

“In the context of a causal model, these potential relationships are represented by the do-calculus [21] $P(Y | do(E))$ and $P(Y | do(P))$, while allowing flexible relationships between gene products.”

To clarify the prior formulation, we have cited references to the original INTACT method as well as Dempster-Shafer theory, which justify and detail how colocalization evidence can be used as a prior for PCG inference. We have included the following sentence in section 2.1:

“Similar to the original INTACT method [22], the incorporation of probabilistic colocalization evidence in the Multi-INTACT model relies on Bayes’ rule and is justified by Dempster’s rule of combination [23].”

Additional comments:

1. In Figure 1 the distinction between solid and dashed edges should be described.

We have added:

“The solid edges represent the graphical model assumed by most multivariable Mendelian randomization methods. The dashed-lines emphasize that Multi-INTACT is designed to be robust to situations in which there are effects between gene products or there are violations of the exclusion restriction (i.e., direct-effect genetic variants).”

to the Figure 1 caption to clarify the distinction between solid and dashed edges.

2. Does the statistical power for the eQTL and pQTL studies affect the ability to distinguish between $E \rightarrow P$ and $P \rightarrow E$?

Multi-INTACT is not designed to infer the presence of effects between gene products (e.g., $E \rightarrow P$ or $P \rightarrow E$). Instead, it is a conditional model focusing on the relationships of $E \rightarrow Y$

and $P \rightarrow Y$. (see also our response to Reviewer 1's Comment 3.) We have added new results (Supplementary Figure S14) to show the statistical power for eQTL and pQTL studies affects the ability to detect PCGs.

3. What is the percent variance explained (population-level PVE) for the simulation studies? How does this compare to the other datasets shown in the Results?

The mean PVE across all simulated complex traits is 0.159 (see section 5.5). The mean (SD) of the estimated heritability across all metabolites (computed by GCTA) is 0.186 (0.149). We have now noted this result in section 5.5.

4. In Figure 3 and the supplementary figures showing "distributions of posteriors for each gene product-to-trait effect scenario", is there something that characterizes simulated loci when the correct posterior type has low posterior probability? Dense LD? Any intuition about what characteristics of loci may lead to lower sensitivity?

This is an excellent question. We find that low posterior probabilities for genuine PCGs are often due to insufficient power in detecting and fine-mapping the underlying molecular QTLs. We have added supplementary Figure S14 to highlight this issue. Factors such as LD patterns and sample size collectively affect the power of molecular QTL mapping, leading to the underestimation of PCG posterior probabilities.

5. For the 30% of annotated KBA pairs that are not found by Multi-INTACT, can the authors detect any patterns (maybe unmodeled mechanisms) that would lead to further method development and higher power? Also for the analysis of the concordance of direction of gene product through expression and protein abundance, are there other mechanisms that might lead to appearance of mis-matching direction?

We appreciate these questions and believe that several factors contribute to the under-detection of KBA pairs in Multi-INTACT analysis. As previously discussed, insufficient power in detecting

pQTL and eQTL is a significant factor. Note that our simulation study is designed to mimic our real data analysis, where we find that the power of Multi-INTACT is approximately 50%. We expect that future molecular QTL data with a higher signal-to-noise ratio will enhance the power of PCG discovery.

It's also possible that some KBA genes are mediated by gene products other than mRNA or protein levels, such as isoform usage. While we have presented Multi-INTACT as a method that accounts for two gene products, it could be extended to include more, potentially increasing its power. We aim to explore this in our future work.

The mismatch in effect size directions is particularly interesting, and we plan to investigate it further in our future work. One possible explanation for the discrepancies in this dataset may be related to the tissue relevance of the molecular QTL data: the metabolite data comes from the METSIM project, which includes only male participants. We observed that tissues with the lowest concordance are female-specific, which could contribute to the mismatch. In contrast, the tissue with the highest concordance is the liver, which is well-known for its central role in metabolism.

6. What background set is used in the gene set analysis (lines 281 onward)?

Our INTACT-GSE analysis does not use a separate test and background set. We form a list of all genes that have represented encoded proteins and RNA transcripts among the UK biobank and GTEx liver datasets, respectively. The full list of genes is included in our linked data repository. It includes 674 unique genes. We test for enrichment of each GO term within this list based on the aggregated probability of putative causality (see section 5.8 for the definition of this probability).

References

- [1] Gusev, A. *et al.* Integrative approaches for large-scale transcriptome-wide association studies. *Nature genetics* **48**, 245–252 (2016).
- [2] Zhu, Z. *et al.* Integration of summary data from gwas and eqtl studies predicts complex trait gene targets. *Nature genetics* **48**, 481–487 (2016).
- [3] Porcu, E. *et al.* Mendelian randomization integrating gwas and eqtl data reveals genetic determinants of complex and clinical traits. *Nature communications* **10**, 1–12 (2019).
- [4] Zhang, Y. *et al.* Ptwas: investigating tissue-relevant causal molecular mechanisms of complex traits using probabilistic twas analysis. *Genome biology* **21**, 1–26 (2020).
- [5] Wang, L., Wen, X. & Morrison, J. Imperfect gold standard gene sets yield inaccurate evaluation of causal gene identification methods. *Communications Biology* **7**, 873 (2024).
- [6] VanderWeele, T. J., Tchetgen, E. J. T., Cornelis, M. & Kraft, P. Methodological challenges in mendelian randomization. *Epidemiology (Cambridge, Mass.)* **25**, 427 (2014).
- [7] Burgess, S., Bowden, J., Fall, T., Ingelsson, E. & Thompson, S. G. Sensitivity analyses for robust causal inference from mendelian randomization analyses with multiple genetic variants. *Epidemiology* **28**, 30–42 (2017).
- [8] Angrist, J. & Imbens, G. Identification and estimation of local average treatment effects. *Econometrica* **62** (1994).
- [9] Angrist, J. D. & Pischke, J.-S. *Mostly harmless econometrics: An empiricist's companion* (Princeton university press, 2009).
- [10] Mancuso, N. *et al.* Probabilistic fine-mapping of transcriptome-wide association studies.

Nature genetics **51**, 675–682 (2019).

- [11] Barfield, R. *et al.* Transcriptome-wide association studies accounting for colocalization using egger regression. *Genetic epidemiology* **42**, 418–433 (2018).
- [12] Hukku, A., Sampson, M. G., Luca, F., Pique-Regi, R. & Wen, X. Analyzing and reconciling colocalization and transcriptome-wide association studies from the perspective of inferential reproducibility. *The American Journal of Human Genetics* **109**, 825–837 (2022).
- [13] Wen, X., Pique-Regi, R. & Luca, F. Integrating molecular qtl data into genome-wide genetic association analysis: Probabilistic assessment of enrichment and colocalization. *PLoS genetics* **13**, e1006646 (2017).
- [14] Hukku, A. *et al.* Probabilistic colocalization of genetic variants from complex and molecular traits: promise and limitations. *The American Journal of Human Genetics* **108**, 25–35 (2021).
- [15] Efron, B., Tibshirani, R., Storey, J. D. & Tusher, V. Empirical bayes analysis of a microarray experiment. *Journal of the American statistical association* **96**, 1151–1160 (2001).
- [16] Kendziorski, C., Newton, M., Lan, H. & Gould, M. On parametric empirical bayes methods for comparing multiple groups using replicated gene expression profiles. *Statistics in medicine* **22**, 3899–3914 (2003).
- [17] Newton, M. A., Noueiry, A., Sarkar, D. & Ahlquist, P. Detecting differential gene expression with a semiparametric hierarchical mixture method. *Biostatistics* **5**, 155–176 (2004).
- [18] Muralidharan, O. An empirical bayes mixture method for effect size and false discovery rate estimation. *The Annals of Applied Statistics* 422–438 (2010).
- [19] Efron, B. *Large-scale inference: empirical Bayes methods for estimation, testing, and prediction*, vol. 1 (Cambridge University Press, 2012).
- [20] Stephens, M. False discovery rates: a new deal. *Biostatistics* **18**, 275–294 (2017).

[21] Pearl, J. & Mackenzie, D. *The book of why: the new science of cause and effect* (Basic books, 2018).

[22] Okamoto, J. *et al.* Probabilistic integration of transcriptome-wide association studies and colocalization analysis identifies key molecular pathways of complex traits. *The American Journal of Human Genetics* **110**, 44–57 (2023).

[23] Dempster, A. P. Upper and Lower Probabilities Induced by a Multivalued Mapping. *The Annals of Mathematical Statistics* **38**, 325 – 339 (1967). URL <https://doi.org/10.1214/aoms/1177698950>.

Second round of review

Reviewer 1

The authors have thoroughly addressed all of my concerns, and I recommend accepting the manuscript.

I do have two additional minor comments that do not require further review from me:

1. The authors mentioned using real genotype data for simulations, which prevented them from varying the sample size. For future work, I recommend considering the TWAS simulation framework developed by Wang et al. (Bioinformatics, 2023). This framework allows for simulating genotypes with varying sample sizes while preserving the original LD patterns using real genotype data as an LD reference.

2. In the first step of Multi-INTACT, the authors stated, “we perform fine-mapping analysis of eQTL, pQTL, and GWAS data.” After reviewing the methods section, it remains unclear to me what is specifically meant by “fine-mapping analysis” in this context. The authors could clarify this in both Figure 1 and line 156 to improve understanding.

As long as the authors revise the manuscript to address comment number 2, no further review from me is needed.

Reviewer 2

I have no additional comments

Reviewer 3

Authors' response to reviewers

Abc